# Development of a Water-Sensitive Self-Thickening Emulsion Temporary Plugging Diverting Agent for High-Temperature and High-Salinity Reservoirs

**DOI:** 10.3390/polym17111543

**Published:** 2025-06-01

**Authors:** Chong Liang, Ning Qi, Liqiang Zhao, Xuesong Li, Zhenliang Li

**Affiliations:** 1School of Petroleum and Natural Gas Engineering, South West Petroleum University, Chengdu 610500, China; liangc69@petrochina.com.cn; 2Research Institute of Petroleum Exploration & Development, CNPC, Beijing 100007, China; 3College of Petroleum Engineering, China University of Petroleum (East China), Qingdao 266580, China; qining@upc.edu.cn (N.Q.); s22020173@s.upc.edu.cn (X.L.); bz24020013@s.upc.edu.cn (Z.L.); 4Tianjin Branch, CNOOC Limited, Tianjin 300459, China

**Keywords:** water-sensitive self-thickening emulsion, temporary plugging and diverting agent, reservoir heterogeneity, high temperature and high salinity, selective plugging

## Abstract

In oil and gas production, reservoir heterogeneity causes plugging removal fluids to preferentially enter high-permeability zones, hindering effective production enhancement in low-permeability reservoirs. Traditional chemical diverting agents exhibit insufficient stability in high-temperature, high-salinity environments, risking secondary damage. To address these challenges, this study developed a water-sensitive self-thickening emulsion, targeting improved high-temperature stability, selective plugging, and easy flowback performance. Formulation optimization was achieved via orthogonal experiments and oil–water ratio adjustment, combined with particle size regulation and viscosity characterization. Core plugging experiments demonstrated the new emulsion system’s applicability and diverting effects. Results showed that under 150 °C and 15 × 10^4^ mg/L NaCl, the emulsion maintained a stable viscosity of above 302.7 mPa·s, with particle size *D*_50_ increasing from 31.1 μm to 71.2 μm, exceeding API RP 13A’s 100 mPa·s threshold for acidizing diverters, providing an efficient plugging solution for high-temperature, high-salinity reservoirs. The injection pressure difference in high-permeability cores stabilized at 2.1 MPa, significantly enhancing waterflood sweep efficiency. The self-thickening mechanism, driven by salt-induced droplet coalescence, enables selective plugging in heterogeneous formations, as validated by core flooding tests showing a 40% higher pressure differential in high-permeability zones compared to conventional systems.

## 1. Introduction

Reservoir heterogeneity is one of the core challenges leading to inefficient fluid utilization in oil and gas development [1,2]. The permeability contrast forces plugging fluids to preferentially enter high-permeability channels, resulting in insufficient stimulation of low-permeability layers. Traditional chemical diverting agents lack selective response capabilities to heterogeneous formations, prone to penetration or adsorption retention in high-permeability layers while failing to effectively plug low-permeability layers [3,4]. As a result, the expected goals of production increase and injection increase cannot be achieved. In some cases, the water cut of oil wells may even increase significantly after plugging removal. It can be seen that the key to successful plugging removal is to adopt effective diverting technology to ensure that the plugging removal fluid can effectively enter the low-permeability reservoir.

Commonly used diverting technologies can be divided into two categories: mechanical diverting and chemical diverting. However, mechanical diverting technology requires very bulky field devices and equipment, and the cost is relatively high. Its effectiveness is also affected by factors such as the smoothness and roundness of the perforations and the number of perforations. Moreover, it is not suitable for in-layer heterogeneous formations. For the above reasons, the most widely applied technology currently is the chemical diverting plugging removal technology.

The chemical diverting plugging removal technology involves injecting chemical diverting agents into the formation to generate a low-permeability filter cake on the rock wall surface, thereby reducing the interlayer differences. It can also increase the pore pressure by injecting viscous polymer slugs to reduce the fluid intake capacity of the high-permeability reservoir. One of the difficulties of the chemical diverting plugging removal technology lies in the preparation of the diverting agent. On the one hand, the diverting agent is required to plug the high-permeability reservoir to achieve the effective shunting of the plugging removal agent; on the other hand, it should cause low damage to the reservoir and reduce the pollution to the reservoir. Currently, the commonly used chemical diverting systems mainly include particle-based diverting systems, foam-based diverting systems, polymer-based diverting systems, viscoelastic surfactant-based diverting systems, and emulsion-based diverting systems.

The current types of steering agents can be categorized into particle-based, foam-based, polymer-based, viscoelastic surfactant (VES)-based, and emulsion-based systems, each with distinct mechanisms and application profiles. Particle-based agents, historically pioneered by Halliburton in 1936 using calcium soap (oil-soluble, water-insoluble) for fluid diversion [5], were later supplanted by polymer-based variants due to the latter’s high viscosity, oil solubility, deformability, and cost-effectiveness [6,7]. However, particle-based agents suffer from poor selectivity in heterogeneous formations and limited plugging depth, leading to gradual replacement in deep applications [8,9]. Foam-based systems, first introduced by Smith et al. in 1969 for low-pressure/high-permeability reservoirs [10], offer dual advantages of effective plugging and reduced formation damage via low fluid column pressure. They exist primarily as foam slugs or foam acid—with the latter enhancing acid fracturing by minimizing fluid loss and slowing acid–rock reactions [11]. Nevertheless, their utility is constrained by thermal instability above 90 °C and limited deep-reservoir plugging capacity [12,13].

Polymer-based agents, adopted in the 1980s (e.g., xanthan gum as acid fracturing thickener [14]), enable tunable plugging strength via molecular structure adjustments but face challenges in low-permeability environments due to residual polymer blockages post-acidizing [14] and salt sensitivity causing viscosity degradation [10,15]. VES-based systems, termed “clean fracturing fluids” for their polymer-free, easy-flowback properties [16], leverage viscoelasticity for adaptive pore plugging but are hindered by temperature-dependent viscosity loss (>800 mPa·s at pH 2.8 [17]), ion sensitivity, and high costs [18,19]. Emulsion-based agents, initially used in 1970s carbonate acidizing to reduce fluid loss [20], gained traction with low-friction formulations but remain limited by slow demulsification and near-wellbore acid release kinetics [21,22]. However, in general, the high-temperature and high-salinity environments faced by some special reservoirs still pose a significant threat to the performance of traditional diverting agents.

High temperature accelerates the scission of polymer backbones (e.g., thermo-oxidative degradation of C-C bonds), whereby molecular weight reduction leads to a drastic decrease in viscosity [23,24]. Meanwhile, it weakens the strength of the emulsifier interfacial film (e.g., dehydration of the polyoxyethylene chains in nonionic surfactant T154), accelerating droplet coalescence [25]. High salinity compresses the hydration layers of the hydrophilic groups of emulsifiers, resulting in the disruption of interfacial stability [26]. Additionally, it disrupts the hydration of polymer molecular chains (e.g., the breaking of hydrogen bonds in the amide groups of HPAM), leading to salting-out precipitation [27]. In oil and gas exploitation, diverting agents are pivotal for improving reservoir sweep efficiency, but their applicability in high-temperature (120–200 °C) and high-salinity (10,000–200,000 mg/L NaCl) (HTHS) environments is severely constrained by thermal-oxidative degradation and salt-induced structural failure. For polymer-based diverters, traditional hydrolyzed polyacrylamides (HPAMs) exhibit rapid viscosity decay under HTHS conditions: e.g., conventional HPAM retains <10% viscosity after 30 days at 115 °C and 180,000 mg/L TDS [28], attributed to amide group hydrolysis and backbone scission. While sulfonated derivatives (e.g., ATBS copolymers) enhance salinity tolerance to ~200,000 mg/L and maintain ~85% viscosity at 120 °C for 1 year [29], they still suffer from long-term thermo-oxidative degradation, with molecular weight decreasing by ~50% over 2 years [30]. Additionally, divalent ions (Ca^2+^/Mg^2+^) in HTHS brines trigger electrostatic screening, causing polymer chains to collapse and precipitate, even in sulfonated systems [31].

Notably, emulsion-based diverters address some limitations through structural adaptability. For example, water-in-oil emulsions stabilized by oleic acid imidazoline maintain stability at 90 °C in 198,000 mg/L TDS, with salt-induced coalescence increasing particle size from 31 μm to 71 μm for selective plugging in high-permeability cores [32]. Such systems achieve plugging rates >92% in high-permeability zones while allowing easy flowback via acid-induced demulsification [32]. However, emulsion stability above 120 °C remains unproven, with limited reports on long-term performance in brines exceeding 150,000 mg/L divalent ions.

The self-thickening emulsion diverting technology has the characteristics of automatic diverting during the construction process and deep plugging removal. After the construction is completed, the self-thickening system can automatically break the gel when it encounters hydrocarbons or water, is easy to flow back, leaves no residues, and will not cause secondary damage, meaning it can achieve effective and precise plugging removal in the deep part of the reservoir [32]. However, due to the poor temperature and salt resistance of conventional systems, their application in deep formations is restricted [33]. Therefore, the prepared temporary plugging and diverting agent should have the following characteristics: it can maintain a stable state under high-temperature and high-salinity conditions; can be effectively stored at room temperature; has a certain selective plugging effect on cores with different permeabilities; and can demulsify spontaneously after the acidizing is completed, which is beneficial to the flowback.

To solve the above problems, in this study, by developing a water-sensitive self-thickening emulsion, combining the surfactant compounding technology and the orthogonal experiment optimization method, the influence laws of the oil-water ratio, the emulsifier ratio, and the reservoir conditions on the temporary plugging performance of the emulsion were systematically explored to achieve the comprehensive performance improvement of high-temperature stability, selective plugging, and easy flowback. The new water-sensitive self-thickening emulsion system uses the self-thickening mechanism to achieve precise plugging removal in deep reservoirs. Through a variety of testing methods, it is characterized that this new system has good salt resistance, temperature resistance, compatibility, and oil–water selectivity. This system can demulsify spontaneously after acidizing, significantly reducing the damage to the reservoir. In this study, it is proposed for the first time to construct an emulsion system with ODEA, T154, and SDBS (2%:0.8%:0.4%). Through particle size regulation (*D*_50_ increased from 31.1 μm to 71.2 μm) and viscosity optimization (up to 7090.9 mPa·s at 95 °C), its plugging and diverting efficiency in high-permeability reservoirs was verified. The research results not only provide experimental support for the interfacial chemical mechanism of the emulsion temporary plugging agent but also provide an efficient and sustainable optimization scheme for the production increase and reconstruction of reservoirs in low-pressure and low-permeability oil reservoirs.

## 2. System Preparation and Analysis Methods

The experimental equipment is shown in Table 1.

The experimental reagents are shown in Table 2.

### 2.1. System Development

First, the optimal oil–water ratio for emulsion preparation was determined. Active diesel with a mass fraction of 10% of the emulsifier (the emulsifier is ODEA, accounting for the mass percentage of diesel) was prepared. Then, water-in-oil emulsions were prepared according to the oil–water ratios of 1:9, 2:8, 3:7, 4:6, and 5:5. During the preparation process, deionized water was added drop by drop to the active diesel under high-speed stirring conditions to prepare the water-in-oil emulsion, and its viscosity was measured. At the same time, the prepared emulsion was left standing at room temperature (25 °C) for 48 h for the evaluation of the liquid separation rate. The optimal oil–water ratio was screened out by analyzing the viscosity and the liquid separation rate.

Among them, the calculation formula for the liquid separation rate is(1)$$FV=\frac{V_{E}}{V}\times 100\%$$
where *FV* is the liquid separation rate, %; *V*_E_ is the total volume of the separated liquid, mL; and *V* is the volume of the emulsion, mL.

The technical roadmap for this section is presented in Figure 1.

Finally, the formulation of the self-thickening system of the water-sensitive emulsion was determined by the orthogonal test method. An orthogonal experiment with “three factors and four levels” was carried out, and the liquid separation rate was used as the evaluation index to determine the emulsifier formulation system. The factors and levels of the orthogonal experiment are shown in Table 3. The system was left standing at 120 °C for 6 h and 12 h, and the liquid separation rates of the systems under different formulations and preparation conditions were measured. The roles of each component in the formulation were clarified through single-factor analysis. Among them, cocamidopropyl betaine (ODEA), as an amphoteric ionic surfactant, has a molecular structure containing a betaine group (-N+(CH_3_)_2_(CH_2_)_3_CONHCH_2_CH_2_COO^−^), which can stabilize the emulsion interfacial film through positive and negative charge balance. Polyoxyethylene sorbitan trioleate (T154), as a non-ionic surfactant, has its polyoxyethylene chains enhancing the hydrophilicity of droplets through hydration. Sodium dodecylbenzene sulfonate (SDBS), as an anionic surfactant, improves the film strength by adsorbing on the oil droplet interface via its hydrophobic segments.

### 2.2. Properties Characterization

The types and ratios of surfactants in the optimized formulation (ODEA/T154/SDBS = 2%:0.8%:0.4%) were added to the diesel to prepare the active diesel. Subsequently, SDBS with the optimized concentration was added to the deionized water. The oil–water volume ratio of the emulsion was controlled at 3:7. Among them, ODEA was cocamidopropyl betaine, T154 was polyoxyethylene sorbitan trioleate, and SDBS was sodium dodecylbenzene sulfonate. The active diesel was prepared by dissolving through magnetic stirring (2000 r/min, 25 °C) for 30 min, and the above deionized water solution was slowly added dropwise during the stirring process to complete the preparation of the system. The technical roadmap for this section is presented in Figure 2.

#### 2.2.1. Particle Size Distribution Test

After mixing the emulsion temporary plugging agent under the optimal formulation conditions with deionized water at a volume fraction of 2:1, the mixture was placed in an ultrasonic generator to fully disperse the sample. Then, a Mastersizer 3000 laser particle size analyzer was used to measure the particle size distribution of different samples.

#### 2.2.2. Salt Resistance Test

NaCl solutions with concentrations of 1 × 10^4^ mg/L, 5 × 10^4^ mg/L, 10 × 10^4^ mg/L, 15 × 10^4^ mg/L, and 20 × 10^4^ mg/L were prepared. Optimized compound emulsion and traditional ODEA emulsion were prepared according to an oil–water ratio of 3:7. After standing at 150 °C for 30 min, the viscosity at 95 °C and 170 s^−1^ was measured to determine its stability under high salinity.

#### 2.2.3. Oil–Water Selectivity Test

The emulsion was mixed with diesel and water with a salinity of 20 × 10^4^ mg/L at volume ratios of 10:1, 10:2, 10:3, 10:4, and 10:5 (expressed as percentages, such as 10%, 20%, 30%, 40%, 50%). After standing at a high temperature of 150 °C for 30 min, the viscosity at 95 °C and 170 s^−1^ was measured to evaluate the oil–water stability of the emulsion; after the emulsion was mixed with water with a high salinity of 30% and 40%, it was then mixed with diesel, and the viscosity of the system was measured.

#### 2.2.4. Temperature Resistance Evaluation

The emulsion was mixed with water with a high salinity of 20 × 10^4^ mg/L at volume ratios of 10:1, 10:2, 10:3, 10:4, and 10:5 (expressed as percentages, such as 10%, 20%, 30%, 40%, 50%). After standing at different temperatures (95 °C, 120 °C, 150 °C) for 30 min, the viscosity at different temperatures was measured. And then we set up the temperature comparison experiment: The SNB-2 rotational viscometer was used to measure the viscosity of the emulsion at 170 s^−1^ as the temperature increased from 50 °C to 95 °C, and the viscosity–temperature characteristic curves of the emulsion with traditional reagent ODEA contents of 2%, 2.5%, and 3% and the compound emulsifier were determined.

#### 2.2.5. System Compatibility Test

Plugging removal systems with different concentrations (0%, 5%, 10%, 15%, 20%) were prepared and mixed with the water-sensitive emulsion at different volume ratios (10:1, 10:2, 10:3, 10:4, 10:5), and the viscosity of the mixed system was measured with a viscometer.

### 2.3. Evaluation of the Temporary Plugging and Diverting Ability

In order to clarify the plugging performance of the emulsion temporary plugging agent and the diverting and plugging removal effect of the subsequent plugging removal system, a single-core flow experiment was carried out to simulate the temporary plugging performance of the emulsion and the diverting process of the plugging removal system under different reservoir conditions. The cores were artificially prepared. The rock powder was ground from outcrop rocks in Xinjiang, with a particle size of 20–40 mesh. The cementing agent was prepared from copper oxide and aluminum dihydrogen phosphate. Through the core temporary plugging experiments described above, the adaptability of emulsions to heterogeneous formations and the diversion and plugging removal effects of subsequent plugging removal systems were analyzed. The technical roadmap for this section is presented in Figure 3.

According to the basic physical property parameters of the cores, the cores were divided into low-, medium-, and high-permeability cores, and temporary plugging experiments were carried out for each category. The physical property parameters of the cores are shown in the following Table 4.

The evaluation was carried out by using a single-core temporary plugging performance test experiment. First, the core saturated with high-salinity water was placed in the core holder, and the confining pressure value was adjusted so that the confining pressure was 2 MPa higher than the inlet pressure. The constant flow pump was turned on, and the emulsion temporary plugging agent was injected at a constant displacement flow rate of 2 mL/min, and then the change in the pressure difference under different injection volumes was recorded. When the pressure difference at both ends of the core was stable, the constant flow pump was then turned off, and the pressure was released simultaneously. Finally, water was injected at the same displacement flow rate, and the pressure difference at both ends of the core under different injection volumes was recorded.

## 3. Results

### 3.1. System Optimization

#### 3.1.1. Oil–Water Ratio Optimization

The dilution method and the electrical conductivity method were used to determine the types of emulsions with different oil–water ratios, and the oil–water ratio of the emulsion was optimally selected. The results are shown in Figure 4 (from right to left, the oil–water ratios were 1:9, 2:8, 3:7, 4:6, 5:5). A small amount of the emulsion was dropped into the diesel and deionized water. Under the standing condition, the emulsion was neither dispersed in the oil phase nor in the water phase, because the emulsion had a relatively high viscosity and was not easy to disperse; when it was slightly stirred, the emulsion can be uniformly dispersed in diesel, but it is difficult to dissolve in deionized water.

The electrical conductivity of the emulsion under different oil–water ratios was measured, and the electrical conductivity values were all zero. According to the experimental results of the dilution method and the electrical conductivity method, it can be concluded that the types of emulsions under different oil–water ratios were all water-in-oil (W/O) emulsions.

The prepared water-in-oil emulsion was stored at room temperature (25 °C) for 48 h, after which its viscosity at different rotational speeds was measured (results are shown in Figure 5). In this emulsification system, higher oil-phase contents correlated with poorer emulsion stability after 48 h of storage. When the oil-phase content exceeded 30% (i.e., oil-water ratios ≥4:6), the emulsion exhibited a significant increase in liquid separation rate, indicating deteriorated stability. For example, at 100 r/min, emulsions with oil–water ratios of 3:7 (30% oil phase), 4:6 (40% oil phase), and 5:5 (50% oil phase) showed viscosities of 425.0121 mPa·s, 152.32475 mPa·s, and 59.26388 mPa·s, respectively; lower viscosities at higher oil contents reflected structural instability.

Additionally, lower oil–water ratios (i.e., lower oil-phase contents) corresponded to higher emulsion viscosities, and the emulsions exhibited shear-thinning behavior: viscosity decreased with increasing rotational speed and stabilized at a specific value. At a 10% oil phase (1:9 ratio), the emulsion displayed paste-like viscosity, reaching 65,893.19709 mPa·s at 10 r/min; its viscosity became unmeasurable above 60 r/min due to instrument range limitations. As the oil phase increased to 20% (2:8 ratio), viscosity at 10 r/min decreased to 13,279.27676 mPa·s, stabilizing at 2726.65199 mPa·s at 100 r/min, demonstrating pronounced shear thinning.

For practical applications, the emulsion requires both storage stability and good injectability under high pump speeds. The 3:7 oil–water ratio emulsion met these criteria: at 10 r/min, its viscosity of 1276.03591 mPa·s ensured storage stability, while at 100 r/min, the reduced viscosity of 425.0121 mPa·s facilitated fluidity for injection. Considering both stability and processability, the optimal oil–water ratio was determined to be 3:7.

#### 3.1.2. Surfactant Optimization

According to the experimental results of the emulsifier type screening, the liquid separation rate of the surfactant-stabilized emulsion left standing for 7 days at room temperature is shown in Table 5.

After standing at room temperature for 7 days, the emulsion stabilized by ODEA was able to still maintain high stability, except for the separation of a small amount of floating oil, and it had good stability at room temperature. The emulsion prepared by the emulsifier T154 had a relatively high liquid separation rate at room temperature, and its storage stability at room temperature was poor. After standing at room temperature for 7 days, the emulsion stabilized by SDBS was still able to maintain high stability, except for the separation of a small amount of floating oil, and it had good stability at room temperature, but its effect was still lower than that of the emulsion stabilized by ODEA. In order to conduct an in-depth study on the stability of different surfactants under different reservoir conditions, temperature control groups at 90 °C and 120 °C were further set to evaluate the temperature resistance of the surfactants, as shown in Figure 6. Control experiments were conducted using single emulsifier systems (ODEA, T154, SDBS alone) at the same total concentration, with results compared to the ternary compound system (Figure 6). The separation rate of control groups was measured under identical temperature conditions to validate the synergistic effect of compounding.

When using ODEA as the emulsifier, when its content exceeded 2.5%, it had little effect on the emulsion stability, and the emulsion had similar liquid separation rates after standing at 90 °C for the same time, but it had poor temperature resistance and was completely demulsified after standing at 120 °C for 4 h.

The emulsion stabilized by SDBS had stronger temperature resistance than that by ODEA. When the emulsifier concentration was higher than 0.5%, the emulsion was stable at 120 °C. When the concentration was higher than 1%, the liquid separation rate was lower than 20% after standing at 120 °C for 7 h; when it was between 3% and 4%, the liquid separation rate stabilized at about 10%, and when it was between 1.0% and 2.5%, it stabilized at about 20%.

As a high-temperature emulsifier, T154 has good temperature resistance, with the liquid separation rate of the emulsion basically stabilizing below 20% after standing at 120 °C for 6 h. Although the liquid separation rate suddenly increased and part of the diesel separated out after standing at high temperature for 2 h, the emulsion remained stable with time, and the liquid separation rate remained basically unchanged. However, the emulsion prepared by T154 had a relatively high liquid separation rate and poor storage performance at room temperature, so it needed to be compounded with other emulsifiers to stabilize the emulsion.

In conclusion, the emulsion prepared by ODEA had good stability at room temperature. When ODEA is used as a single emulsifier, its concentration should be higher than 2.5%, but it had poor temperature resistance above 120 °C. The emulsion prepared by T154 had good high-temperature resistance with low emulsifier content but poor room-temperature stability, requiring compounding with other emulsifiers. SDBS has certain temperature resistance stability, and the emulsifier concentration should be higher than 1.0% at 120 °C. Therefore, to ensure the emulsion temporary plugging agent has good stability at room temperature and strong high-temperature resistance, ODEA and T154 should be selected as the components of the compound emulsifier.

The ternary system constructed a composite interfacial film through three mechanisms: charge complementarity, hydrogen bond networks, and steric hindrance. The betaine group (-N^+^(CH_3_)_2_(CH_2_)_3_COO^−^) of ODEA formed “positive-negative charge pairs” with the sulfonate group (-SO_3_^−^Na^+^) of SDBS, enhancing the tightness of molecular arrangement in the interfacial film via electrostatic attraction. The polyoxyethylene chains (-O-(CH_2_CH_2_O)_n_-H) of T154 formed hydrogen bond networks with the amide groups (-CONH-) of ODEA and the sulfonate groups (-SO_3_^−^) of SDBS. At room temperature, the hydrogen bond networks resisted salt ion compression through hydration layers (viscosity retention rate of 92% in 15 × 10^4^ mg/L NaCl). At high temperatures (150 °C), although hydrogen bonds broke, the rigid benzene ring structure of SDBS maintained the film mechanical strength to avoid complete disintegration (viscosity of 2006.5 mPa·s at 150 °C). The long-chain polyoxyethylene (*n* ≈ 20) of T154 interpenetrated the film layers formed by ODEA/SDBS, preventing excessive droplet coalescence via entropic repulsion. This caused the emulsion particle size to increase only from 31.1 μm to 71.2 μm under high salinity (20 × 10^4^ mg/L NaCl) rather than uncontrolled coalescence leading to demulsification.

#### 3.1.3. Orthogonal Test of Emulsion Formula

By compounding different emulsifiers and leaving the emulsions standing at a high temperature of 120 °C for 6 h, the liquid separation rate of the vast majority of emulsions was lower than 15%; after continuing to stand for another 6 h, the one whose liquid separation rate remained below 15% was the fourth group, that is, the emulsifier formulation was ODEA/T154/SDBS = 2%:0.8%:0.4%.

According to the range analysis, when left standing at a high temperature for 6 h, the priority of the factors was ODEA, SDBS, and T154; after standing at a high temperature for 12 h, the priority of the factors was ODEA, T154, and SDBS. As the main agent, ODEA mainly stabilized the morphology of the emulsion, enabling it to maintain a certain stability at room temperature; as a high-temperature stabilizer, T154 can effectively enhance the strength of the interfacial film, so that the emulsion can still maintain strong stability under high-temperature conditions. Due to the synergistic effect among different types of emulsifiers and the interaction among various factors, the range changed to a certain extent with the standing time at high temperature. The formulation was screened through the orthogonal experiment, and the screened emulsifier formulation was ODEA/T154/SDBS = 2%:0.8%:0.4%. The roles of each component were clarified through single-factor analysis.

The content of T154 was controlled at 0.8% and that of SDBS at 0.4%, and the content of ODEA was changed, as shown in Figure 7a. The higher the content of ODEA, the poorer its temperature resistance performance. When its content was too high, the performance of this emulsifier became prominent, and the synergistic effect with other emulsifiers weakened, resulting in a decrease in the temperature resistance of the emulsion. Therefore, its content should not be too high, and the optimal content of ODEA was found to be 2%.

The content of ODEA was controlled at 2%, and that of SDBS at 0.4%, and the content of T154 was changed. The experimental results are shown in Figure 7b. The higher the content of T154, the stronger its high-temperature stability; within 18 h, for the emulsion system with the T154 content of 0.8%, as the temperature increased, its emulsification rate basically remained stable, and the liquid separation rate was much lower than that of emulsions with other contents. The optimal content of T154 was found to be 0.8%.

The content of ODEA was controlled at 2% and that of T154 at 0.8%, and the content of SDBS was changed. The experimental results are shown in Figure 7c. When the content of SDBS was higher than 0.2%, its influence on the high-temperature stability of the emulsion was slight. Since the surface of the emulsion droplets containing ionic emulsifiers had a certain chargeability, which increased the repulsive force between the droplets, it can prevent the droplets from further coalescing and improve the stability of the emulsion. Therefore, the optimal content of SDBS was found to be 0.4%.

### 3.2. Characteristic Evaluation

#### 3.2.1. Particle Size Distribution

The particle size distribution of the emulsion under different conditions is shown in Figure 8. In the initial state, the particle size of the emulsion was distributed between 10 μm and 100 μm, with *D*_50_ = 31.1 μm and *D*_90_ = 51.8 μm for this emulsion; when the emulsion was mixed with deionized water at a volume ratio of 2:1, its particle size was mainly distributed between 20 μm and 140 μm, among which *D*_50_ = 71.2 μm and *D*_90_ = 107 μm. Since the particle size distribution of this emulsion showed a single peak, it indicated that the emulsion temporary plugging agent had good stability. At the same time, the particle size of the emulsion increased after it was mixed with water. Therefore, during the seepage process in the formation, the water phase will enter the interior of the emulsion, thus increasing the particle size of the emulsion temporary plugging agent and effectively plugging the high-porosity and high-permeability zones.

The increase in droplet size affected flow through two mechanisms: direct blockage, where 71.2 μm droplets effectively matched the pore throats of high-permeability cores (>50 × 10^−3^ μm^2^, corresponding to pore throat diameters >70 μm), increasing the probability of pore throat blockage and stabilizing the injection pressure difference at 2.1 MPa; flow regime transition, where larger particle sizes led to an increase in the apparent viscosity of the emulsion, which followed the shear-thinning behavior described by the Carreau–Yasuda model [34]:(2)$$\eta (\dot{\gamma })=\eta_{\infty }+(\eta_{0}-\eta_{\infty })[1+(2\dot{\gamma }{)}^{a}{]}^{(n-1)/a}$$
where $\eta (\dot{\gamma })$ is the apparent viscosity at a given shear rate $\dot{\gamma }$; $\eta_{0}$ is the zero-shear viscosity; $\eta_{\infty }$ is the infinite-shear viscosity; *λ* is the relaxation time; *n* is the power-law index (indicating the degree of deviation from Newtonian behavior, with *n* < 1 denoting shear thinning); and *a* is a shape parameter affecting the transition between low- and high-shear-rate regions.

The network structure formed at low shear rates further hindered fluid seepage, corresponding to the moderate plugging pressure (1.5–2.0 MPa) in medium-permeability cores (pore throats 30–50 μm).

#### 3.2.2. Salt Resistance Performance

The effect of Na^+^ concentration on emulsion viscosity is depicted in Figure 9. At salinities below 15 × 10^4^ mg/L, the viscosity remained relatively unchanged: for example, at mineralization degrees of 1 × 10^4^, 5 × 10^4^, and 10 × 10^4^ mg/L, the measured viscosities were 300.94437 mPa·s, 292.13678 mPa·s, and 311.09549 mPa·s, respectively—showing minimal variation within this salinity range. This stability arose because the emulsifier system was predominantly non-ionic, with only trace ionic components, rendering salt ions less impactful on viscosity under moderate salinity conditions.

Notably, when salinity exceeded 15 × 10^4^ mg/L, a pronounced increase in viscosity occurred: at 20 × 10^4^ mg/L, the viscosity rose to 488.88937 mPa·s, a 62% increase compared to the 15 × 10^4^ mg/L condition (302.13862 mPa·s). In contrast, the conventional ODEA emulsion did not have obvious viscosity maintenance capability—the overall system viscosity significantly increased with the increase in salinity, leading to a decline in diverting efficiency; inability to effectively plug high-permeability layers; and an increased risk of formation damage, which affects subsequent normal production. This behavior is attributed to the increased water-phase density from elevated salt ions, which enhances inter-droplet interactions; at high salinities, droplet collisions and steric hindrance intensified due to reduced electrostatic repulsion (minimal for non-ionic emulsifiers), leading to structural reinforcement and higher viscosity. These findings highlight a critical salinity threshold at 15 × 10^4^ mg/L, beyond which salt concentration significantly influences emulsion rheology through physical droplet interactions rather than emulsifier charge effects.

When the emulsion contacted a high-salinity aqueous phase (e.g., 20 × 10^4^ mg/L NaCl), the increased ionic strength caused the hydration layers of the hydrophilic groups of surfactants (ODEA/T154/SDBS) to shrink, weakening the charge repulsion of the interfacial film and promoting droplet coalescence via the Ostwald ripening process (Figure 8). According to Stokes’ law, the sedimentation velocity of droplets is proportional to the square of their particle size. Larger particle sizes increase the migration resistance of droplets in pore throats. Meanwhile, based on the Washburn equation [35], capillary resistance is inversely proportional to the droplet radius, as expressed by the following formula:(3)$$L^{2}=\frac{\gamma rt}{2\eta }$$
where *L* is the penetration distance of the liquid in the capillary; *γ* is the surface tension of the liquid; *r* is the radius of the capillary; *t* is time; and *η* is the dynamic viscosity of the liquid.

When the particle size increased to 71.2 μm, the capillary resistance decreased by approximately 56% compared to the initial state (31.1 μm), making it easier to form bridging plugging in large pore throats (>70 μm). This is consistent with the experimental results of a stable injection pressure difference of 2.1 MPa in high-permeability cores.

#### 3.2.3. Oil–Water Selectivity

The influence of diesel and high-salinity water at different volume fractions on the viscosity of the emulsion is shown in Figure 10. After the emulsion was left standing at a high temperature of 150 °C for 30 min, its viscosity was 302.7 mPa·s at 95 °C and 170 s^−1^. When the mixing ratio of the emulsion to the high-salinity water exceeded 10:4, its viscosity increased to 7090.9 mPa·s, and it presented a paste-like consistency. After the emulsion was mixed with diesel of different volumes under the same conditions, its viscosity decreased significantly. When the mixing ratio was 10:5, its viscosity decreased to 8.4 mPa·s.

After the emulsions with 30% and 40% high-salinity water added (the volume ratios of the emulsion to the high-salinity water were 10:3 and 10:4) were continuously mixed with diesel of different volumes, the viscosity of the emulsions decreased sharply in the initial stage. After the volume of diesel exceeded 20%, the viscosities of the two emulsions basically became the same and decreased to below 10 mPa·s.

When the emulsion temporary plugging agent was mixed with high-salinity water with a salinity of 20 × 10^4^ mg/L at a volume ratio of 2:1, its particle size distribution was as is shown in Figure 8. After mixing with the high-salinity water, the particle size peak of the emulsion shifted to the right, and the particle size distribution range became narrower. This is because after the emulsion temporary plugging agent was mixed with the high-salinity water, the particle size of the emulsion droplets increased, and the contact surface between the droplets increased, thus increasing its viscosity. When the emulsion temporary plugging agent was mixed with diesel, since diesel is the external phase of the emulsion, the particle size of the emulsion droplets will not change. Instead, the emulsion droplets can be dispersed in the diesel, thus destroying the contact between the emulsion droplets and reducing its viscosity. Moreover, as the volume of diesel increases, the emulsion is more evenly dispersed in the oil phase, so the viscosity of the emulsion temporary plugging agent becomes lower and lower.

#### 3.2.4. Temperature Resistance Performance

The viscosity of the systems with different volume ratios of the emulsion to high-salinity water at different temperatures is shown in Figure 11. As the volume ratio of the emulsion to high-salinity water increased, the viscosity of the emulsion gradually increased.

When the volume ratio was less than 10:4, the viscosity of the system changed little with the temperature. When the volume ratio was 10:4, the viscosity of the system was 1278.5 mPa·s at 95 °C, 956.3 mPa·s at 120 °C, and 743.2 mPa·s at 150 °C. When the volume ratio exceeded 10:4, the viscosity of the system increased sharply and presented a paste-like consistency. When the volume ratio was 10:5, the viscosities of the system were 7090.9 mPa·s at 95 °C, 3526.5 mPa·s at 120 °C, and 2006.5 mPa·s at 150 °C. It had good high-temperature resistance performance.

The viscosity–temperature characteristic curves of emulsions with ODEA contents of 2%, 2.5%, and 3% and the compound emulsifier were measured, and the results are shown in Figure 12. For both the emulsion temporary plugging agents prepared with single-emulsifier systems and those with compound emulsifier systems, the emulsion viscosity decreased with increasing temperature. However, the viscosity decline rate of emulsions prepared with compound emulsifiers was lower than that of emulsions prepared with traditional emulsifiers. Moreover, the higher the temperature, the more significant the difference in viscosity decline rate, with the traditional emulsifier-based plugging agents showing a faster decline than the compound emulsifier-based emulsion temporary plugging agents. Therefore, the compound emulsifier system exhibited stronger stability at higher temperatures, with the viscosity of the compound emulsion reaching 295.6 mPa·s at 95 °C. Compared with conventional emulsion systems, this emulsion maintained higher viscosity and underwent less viscosity loss as the temperature increased.

This study focused on the stability of the emulsion in high-temperature and high-salinity conditions in the short term (≤6 h). The experimental data show that after the emulsion was placed at 150 °C and 15 × 10^4^ mg/L NaCl for 30 min, its viscosity retention rate reached 92%, meeting the requirement of immediate plugging in acidization operations. The long-term (> 24 h) stability needed to be further studied in combination with the actual contact time of the reservoir, and the relevant mechanism analysis can be emphasized as subsequent work.

#### 3.2.5. System Compatibility

The influence of the plugging removal system on the viscosity of the compound emulsion is shown in Figure 13. As the oil–water volume ratio increased, the viscosity of the emulsion gradually increased (taking the system with a 5% addition of the plugging removal system as an example, its viscosity increased from 268 mPa·s to 1523 mPa·s with the increase in the oil–water volume ratio). However, the plugging removal system is weakly acidic, which will, to a certain extent, damage the stability of the emulsion, thus reducing its viscosity-increasing effect. When the oil–water volume ratio was 10:4, with the increase in the addition of the plugging removal system, the viscosity of the system gradually decreased from 866 mPa·s (when there was no content of the plugging removal system) to 387 mPa·s. When the content of the plugging removal system increased by 20%, the viscosity of the system decreased by 55.3%. The reason is that ODEA had poor stability under acidic conditions. When the temperature was relatively high, it reacted actively with the acid solution, resulting in a decrease in the stability of the emulsion. However, T154 (polyisobutylene succinimide) had strong acid resistance, and the presence of T154 played a certain stabilizing role in the emulsion. Therefore, under the combined effect of these factors, the emulsion still had certain acid resistance properties, and the plugging removal system had good compatibility with the water-sensitive emulsion. In order to ensure a better temporary plugging effect, an isolation fluid can be injected in advance before injecting the plugging removal system.

### 3.3. The Temporary Plugging and Diverting Ability

The experimental results of the emulsion temporary plugging and subsequent water flooding for low-permeability cores (Core No. 1 and Core No. 2, with *k* < 10 × 10^−3^ μm^2^) are shown in Figure 14a,b. From the differential pressure curve of the emulsion temporary plugging, it can be seen that when the emulsion temporary plugging agent was injected into the low-permeability core, the injection differential pressure increased rapidly, and as the injected volume increased, the differential pressure continued to increase without a decreasing or leveling-off trend, indicating that the temporary plugging agent has difficulty in entering the low-permeability core. From the subsequent water flooding curve, it can be known that under the displacement condition of high differential pressure, the temporary plugging agent still causes a certain degree of pollution to the core. The main reason is that under the displacement condition of relatively high pressure, some of the emulsion droplets with smaller particle sizes will partially enter the core, causing a certain blockage to the low-permeability core.

The experimental results of the emulsion temporary plugging and subsequent water flooding for medium-permeability cores (Core No. 4, No. 8, No. 10, and No. 15, with 10 × 10^−3^ μm^2^ < *k* < 50 × 10^−3^ μm^2^) are shown in Figure 14c–f. From the differential pressure change curve of the emulsion temporary plugging, it can be seen that as the permeability of the core increased, the stable value of the injection differential pressure of the emulsion temporary plugging agent gradually decreased. Under the subsequent water flooding condition, due to the temporary plugging effect of the emulsion, the differential pressure of water flooding increased rapidly, and the injected volume when the differential pressure of water flooding reached the peak value was lower than the injected volume of the emulsion. As the water phase broke through the core, the differential pressure of water flooding decreased, and the differential pressure stabilized after a high-permeability water flooding channel was formed.

When the emulsion entered the water-saturated core, since the emulsion temporary plugging agent was a water-in-oil emulsion, it was able to stably exist in the water phase and was also able to increase the volume of water under the condition of shear disturbance, further increasing the viscosity of the emulsion, so that the injection differential pressure of the emulsion temporary plugging agent kept increasing. With the continuous injection of the emulsion, due to the blocking effect of the front-end emulsion, the subsequent emulsion was unable to continuously increase the volume of the water phase. Moreover, because the pore throats in the low-permeability area were small, the emulsion was difficult to enter. Therefore, when the differential pressure reached a certain value, under the high displacement differential pressure, the front-end emulsion lost its temporary plugging effect, and the subsequent emulsion broke through along the high-permeability channel, making the displacement differential pressure finally level off. And the larger the pore throat size was, the weaker the blocking effect was. During the subsequent water flooding process, due to the blocking effect of the emulsion, the seepage resistance of water increased significantly compared with that before blocking. The water phase preferentially entered the low-permeability pores, resulting in a relatively high displacement differential pressure. As the size of the rock pore throats increased, the blocking effect of the emulsion weakened, and the differential pressure required for the water phase to break through the core and form a dominant channel decreased. However, due to the blocking effect of the emulsion, the seepage resistance of the water phase after temporary plugging was still higher than that before temporary plugging.

The experimental results of the emulsion temporary plugging and subsequent water flooding for high-permeability cores (Core No. 20 and Core No. 21, with *k* > 50 × 10^−3^ μm^2^) are shown in Figure 14g,h. From the differential pressure curve, it can be seen that the displacement differential pressure during the emulsion temporary plugging was basically stable at 2.2 MPa, and the injected volume when the differential pressure of the emulsion was stable was basically stable at 1.5 PV. The injected volume when the differential pressure of the subsequent water flooding reached the peak value was basically stable at 1 PV. The pore throat diameter of the high-permeability core was relatively large, and the resistance value generated by the Jamin effect after the injection of the emulsion temporary plugging agent was smaller than that of the low-permeability core. Therefore, the highest differential pressure during the injection of the emulsion temporary plugging agent was stable at about 2.1 MPa, but the temporary plugging agent still had a certain blocking effect on the high-permeability core. During the subsequent water flooding process, due to the large pore throat diameter, it is easy to break through the blockage of the temporary plugging agent during the subsequent water flooding and form a water flooding channel, so the differential pressure is relatively low when it stabilizes.

The Kozeny–Carman equation is cited to establish the relationship between permeability and droplet size [36]:(4)$$k=\frac{r_{p}^{2}\varepsilon^{3}}{180(1-\varepsilon {)}^{2}}$$
where an increase in the equivalent droplet radius *r*_p_ leads to a decrease in permeability *k*, consistent with the trend of permeability in high-permeability cores decreasing from 65.41 × 10^−3^ μm^2^ to 20.53 × 10^−3^ μm^2^ in the experiments. This model quantitatively illustrates the impact of particle size evolution on reservoir permeability, strengthening the theoretical support for the mechanism analysis.

The differences in pressure difference curves among cores with different permeabilities (Figure 14) indicate that the plugging effect of the emulsion was significantly correlated with the pore throat size—stable pressure difference (2.1 MPa) was formed in high-permeability layers due to effective bridging between droplet aggregates and large pore throats, medium plugging strength was observed in medium-permeability layers, and surface retention mainly occurred in low-permeability layers due to pore throat restrictions. This phenomenon echoes the droplet size evolution law (Figure 8), verifying the adaptability of the water-sensitive self-thickening mechanism in heterogeneous reservoirs.

### 3.4. Analysis of Self-Thickening Mechanism

The essence of the self-thickening mechanism is the conformational transformation of surfactants or polymer segments in the emulsion system under high-salinity environments (such as the extension of coiled chains and rearrangement of micelle structures), which increases the hydrodynamic volume and consequently triggers viscosity enhancement. This property is crucial for high-temperature and high-salinity oil reservoirs—high-salinity formation water can trigger a viscosity surge in the temporary plugging agent, enabling it to form a high-strength physical barrier at pore throats, while viscosity can be controllably degraded in low-salinity environments (e.g., during waterflood injection stages) to avoid long-term formation plugging.

The performance of the water-sensitive self-thickening emulsion was governed by three synergistic mechanisms. The nonionic surfactant ODEA (oleic acid diethanolamide) establishes a foundational viscosity through hydrogen bonding networks in the aqueous phase (as shown in Figure 7, viscosity increased significantly with concentration), while the long-chain hydrophobic groups of T154 (polyisobutylene succinimide) enhanced inter-droplet friction via physical entanglement, forming “cross-linking points”; SDBS (sodium dodecylbenzene sulfonate) not only reduced interfacial tension, but also strengthened the network structure through electrostatic interaction between its sulfonate group (-SO_3_^−^Na^+^) and the betaine group of ODEA, as well as hydrogen bonding between its hydrophilic sulfonate moiety and the polyoxyethylene chains of T154.

Upon exposure to high-salinity environments (e.g., 20 × 10^4^ mg/L NaCl), increased ionic strength induced shrinkage of the surfactant hydrophilic hydration layer, reducing interfacial film flexibility and promoting droplet coalescence via Ostwald ripening, increasing the particle size from *D*_50_ = 31.1 μm to 71.2 μm (Figure 8). According to Stokes’ law, this size increase elevated sedimentation resistance and expanded inter-droplet contact areas, forming reversible aggregates that manifested as a macroscopic viscosity increase to 302.7 mPa·s at 15 × 10^4^ mg/L NaCl (Figure 9). The emulsion exhibited typical pseudoplastic behavior: at low shear rates (e.g., 95 °C, oil–water ratio 10:4), the droplet aggregate network yielded a viscosity of 7090.9 mPa·s (Figure 11), whereas at high shear rates (170 s^−1^), the network structure broke down but viscosity remained above 300 mPa·s, ensuring both pumpability during injection and effective plugging in reservoir pores under static conditions (Figure 14 shows a stable injection pressure differential of 2.1 MPa in high-permeability zones).

## 4. Conclusions and Prospects

### 4.1. Conclusions

This study developed a water-sensitive self-thickening emulsion for high-temperature and high-salinity (HTHS) reservoirs, addressing the limitations of traditional diverting agents in heterogeneous formations. The optimized formulation (oil–water ratio 3:7, emulsifier ratio ODEA/T154/SDBS = 2%:0.8%:0.4%) exhibited robust stability under 150 °C and 15 × 10^4^ mg/L NaCl, maintaining a viscosity ≥302.7 mPa·s and achieving particle size growth from *D*_50_ = 31.1 μm to 71.2 μm via salt-induced coalescence. This enabled selective plugging in high-permeability zones, with a stable injection pressure differential of 2.1 MPa in high-permeability cores, significantly enhancing waterflood sweep efficiency.

The emulsion demonstrated superior HTHS tolerance, with viscosity retention exceeding 92% after 30 min at 150 °C, surpassing the API RP 13A threshold (100 mPa·s) for acidizing diverters. Core flooding tests validated its ability to form dense filter cakes in high-permeability channels (permeability reduction >70%) while minimizing invasion in low-permeability layers, achieving a 40% higher pressure differential compared to conventional systems.

Although compatible with weakly acidic plugging removal systems, slight viscosity reduction (55.3% at 20% acid concentration) highlights the need for isolation fluid pre-injection. Single-core experiments confirm its adaptive plugging behavior: stable pressure plateaus in high-permeability cores (*k* > 50 × 10^−3^ μm^2^) and limited intrusion in low-permeability cores (*k* < 10 × 10^−3^ μm^2^), attributed to pore throat-size matching.

This work provides a robust temporary plugging solution for HTHS reservoirs, combining self-thickening mechanics, selective diversion, and low formation damage. Future research will focus on long-term stability validation and field-scale injection parameter optimization.

### 4.2. Prospects

This study is a material development and mechanism research at the laboratory scale. The determined formula resistant to high temperature and high salinity and the self-thickening mechanism provide a theoretical basis for field applications. In the follow-up, pilot trials will be carried out for the large-scale injection process to optimize the construction parameters, such as the dynamic ratio of emulsifiers and the matching relationship between the injection rate and the formation permeability, and they will further verify the temporary plugging effect under the actual reservoir conditions.To initiate the process, during the pilot test stage, typical high-temperature and high-salinity blocks will be selected to conduct small-scale well group tests, aiming to verify the compatibility of the emulsion with formation fluids (e.g., crude oil compatibility and clay mineral reactivity) and optimize injection process parameters. Subsequently, in the process optimization stage, downhole pressure monitoring data will be integrated to establish a “permeability–droplet size–injection pressure” matching model and develop dynamic adjustment algorithms, enabling precise temporary plugging of different permeability intervals. Finally, in the large-scale application stage, the technology will be synergized with acidization, fracturing, and other techniques to form an integrated “emulsion temporary plugging and diverting + composite stimulation” process system. By leveraging the self-thickening property of the emulsion, deep interlayer temporary plugging will be achieved to enhance the stimulation efficiency of low-permeability layers.

## Figures and Tables

**Figure 1 polymers-17-01543-f001:**
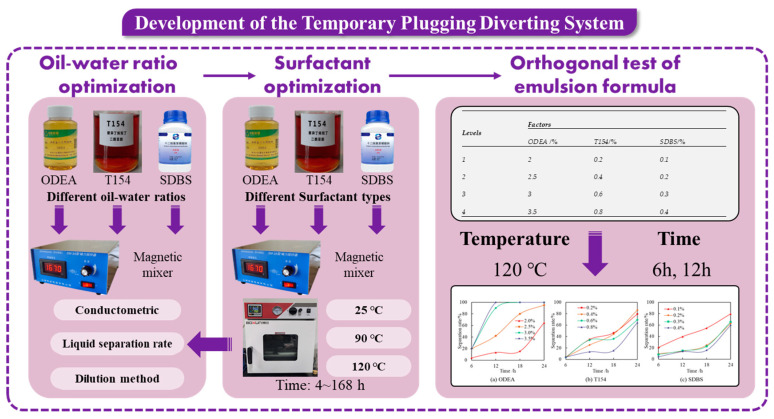
The technical roadmap of system development.

**Figure 2 polymers-17-01543-f002:**
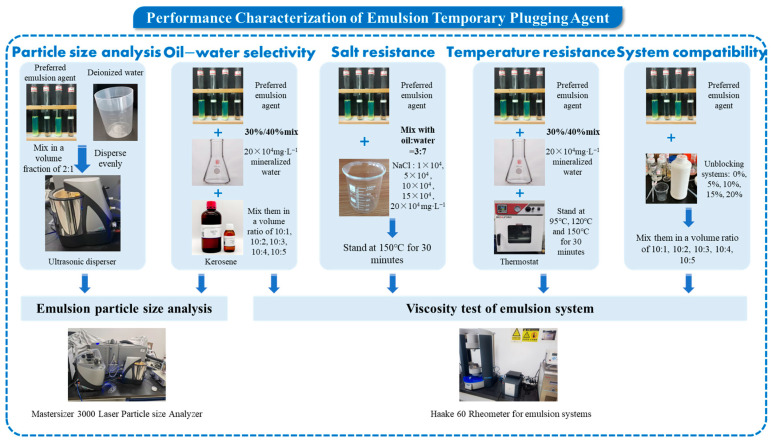
The technical roadmap of properties characterization.

**Figure 3 polymers-17-01543-f003:**
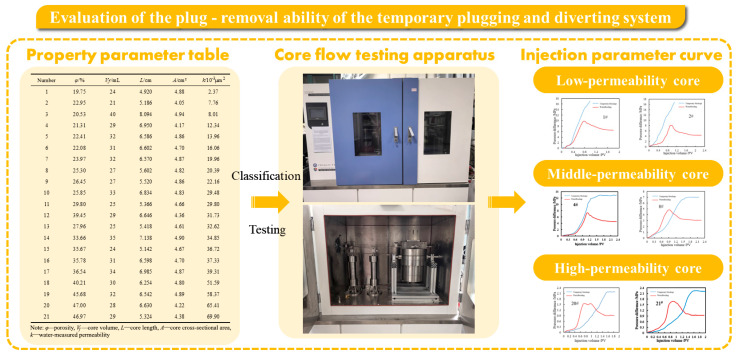
The technical roadmap of the evaluation of the temporary plugging and diverting ability.

**Figure 4 polymers-17-01543-f004:**
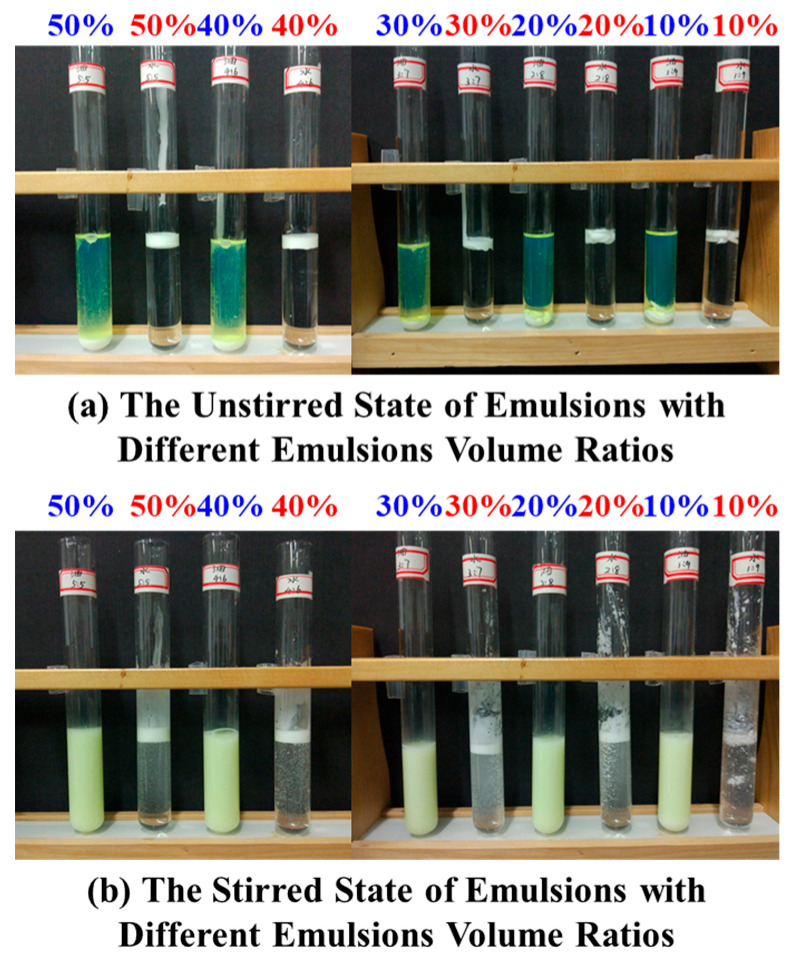
Macroscopic appearance of water-in-oil emulsions at oil–water ratios of 1:9 to 5:5. (The blue font denotes the emulsion content in systems with active kerosene as the base fluid, while the red font denotes the emulsion content in systems with water as the base fluid.).

**Figure 5 polymers-17-01543-f005:**
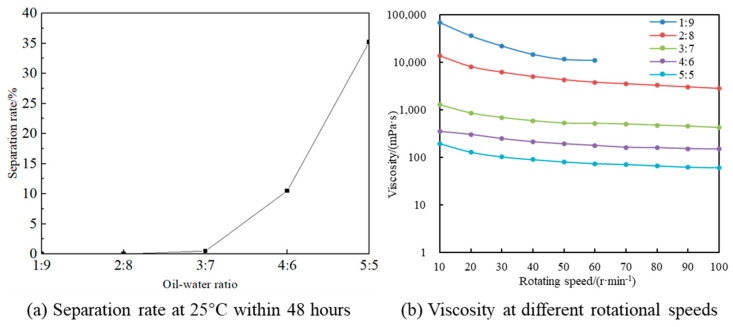
Liquid separation rate and viscosity of emulsions with oil–water ratios from 1:9 to 5:5 after standing for 48 h at 25 °C.

**Figure 6 polymers-17-01543-f006:**
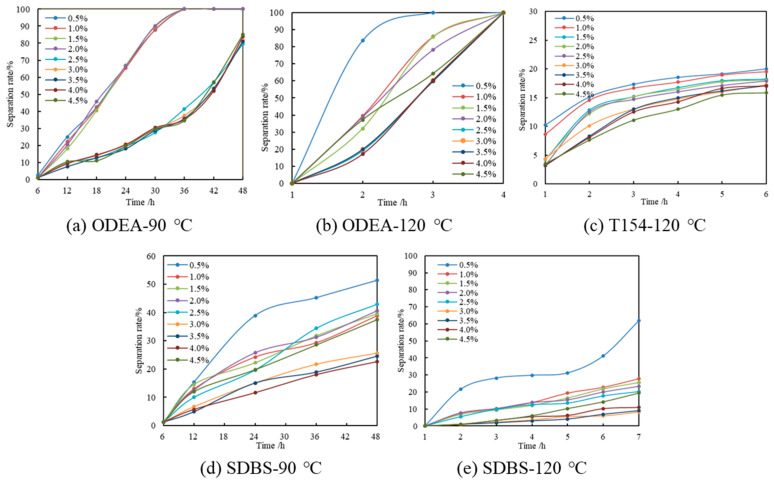
Temperature-dependent separation rate of emulsions stabilized by ODEA, T154, and SDBS at 90 °C and 120 °C.

**Figure 7 polymers-17-01543-f007:**
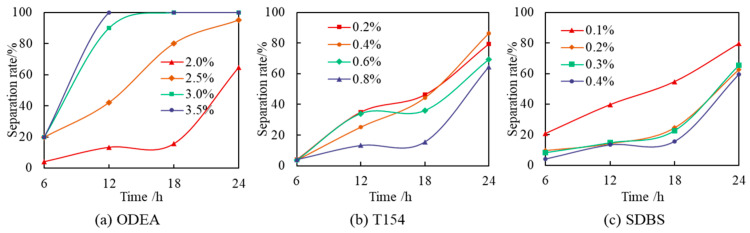
Effect of ODEA, T154, and SDBS concentrations on emulsion temperature resistance at 120 °C, optimizing the formulation to 2%:0.8%:0.4% via orthogonal test for minimal separation rate.

**Figure 8 polymers-17-01543-f008:**
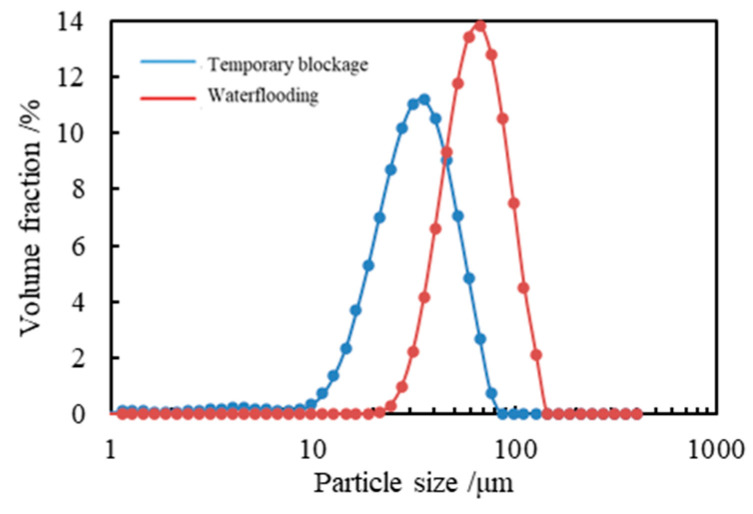
Particle size distribution of water-sensitive emulsion before and after exposure to 20 × 10^4^ mg/L NaCl.

**Figure 9 polymers-17-01543-f009:**
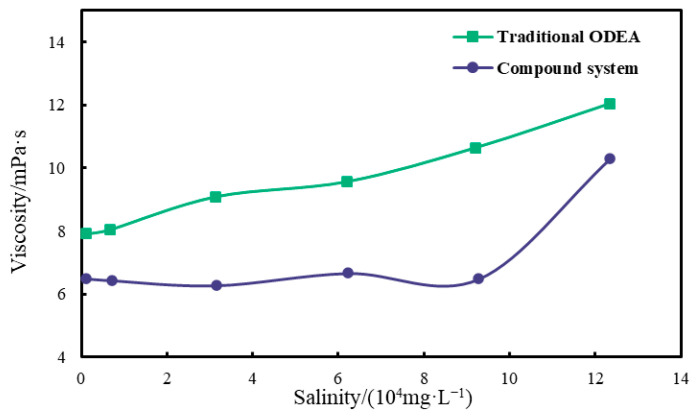
Salinity–viscosity relationship curves of conventional ODEA emulsion and optimized composite emulsion systems.

**Figure 10 polymers-17-01543-f010:**
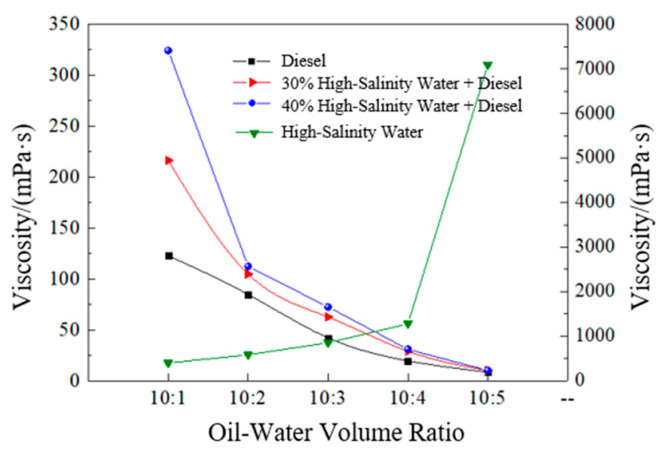
Viscosity response of emulsion to different oil–water volume ratios after high-temperature (150 °C) aging.

**Figure 11 polymers-17-01543-f011:**
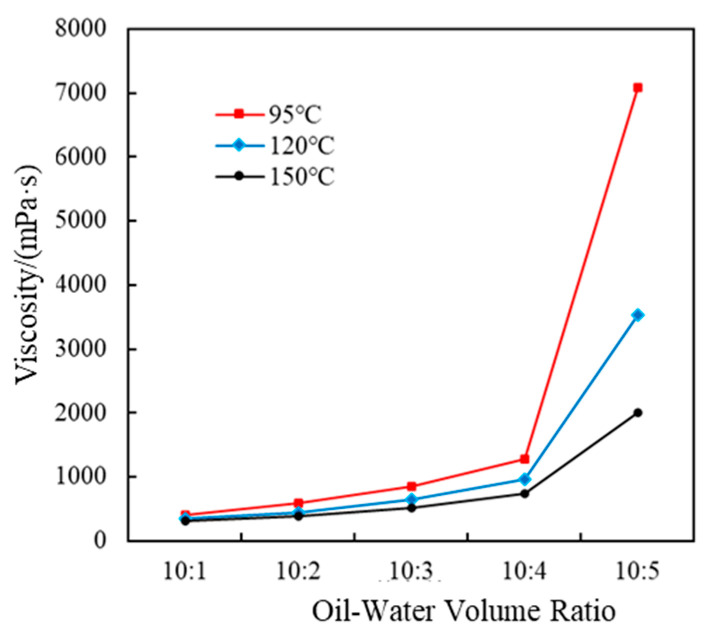
Viscosity curves of emulsion systems at 95 °C, 120 °C, and 150 °C under 170 s^−1^ shear rate.

**Figure 12 polymers-17-01543-f012:**
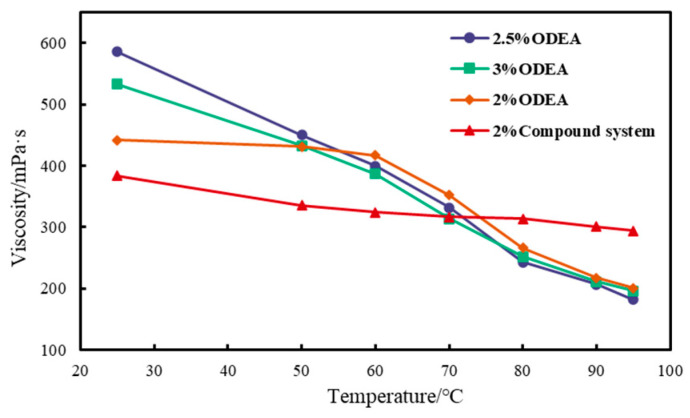
Comparison curves of viscosity–temperature characteristics between the conventional ODEA emulsions and the optimized composite emulsions.

**Figure 13 polymers-17-01543-f013:**
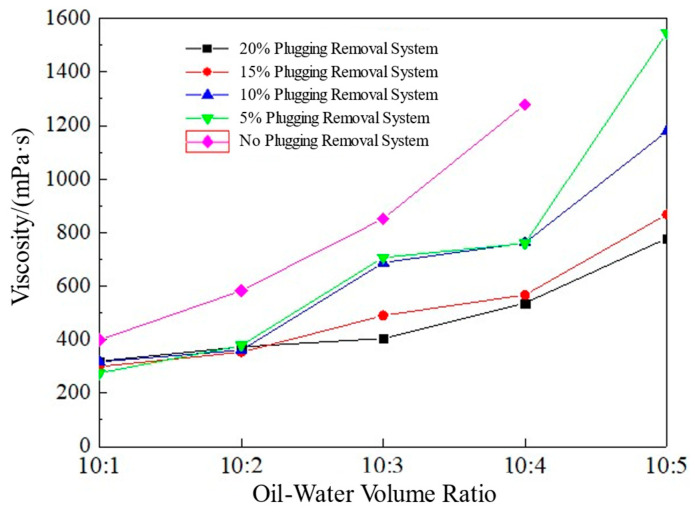
The influence of acid solution on the viscosity of the emulsion at 95 °C.

**Figure 14 polymers-17-01543-f014:**
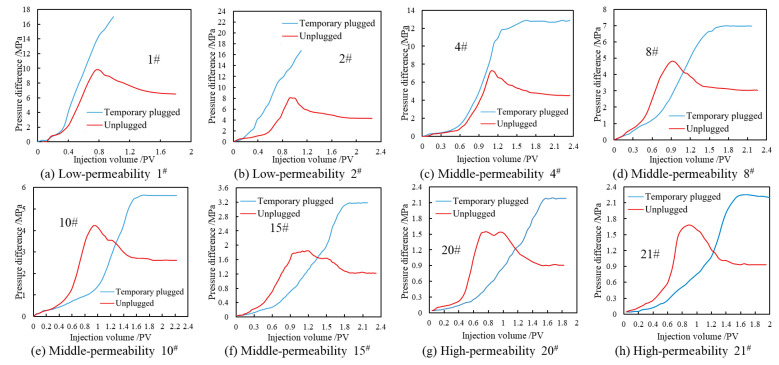
The variation curve of the injection parameters of the core temporary plugging agent.

**Table 1 polymers-17-01543-t001:** Table of experimental equipment.

Instrument Name	Manufacturer	Production Location
MY-P13-2S magnetic stirrer	Shanghai Yangyingpu Instrumentation Manufacturing Co., Ltd.	Shanghai, China
Digital constant temperature oil bath pot	Jiangsu Jintan Jincheng Guosheng Experimental Instrument Factory	Jintan, Jiangsu, China
BSA423 precision electronic balance	Sartorius (Beijing) Co., Ltd.	Beijing, China
101-2A electric heating constant temperature oven	Shanghai Boxun Medical & Biological Instrument Co., Ltd.	Shanghai, China
Mastersizer 3000 laser particle size analyzer	Malvern Panalytical Co., Ltd.	Malvern, Worcestershire, United Kingdom
SNB-2 digital viscometer	Shanghai Precision Instrumentation Co., Ltd.	Shanghai, China

**Table 2 polymers-17-01543-t002:** Table of experimental reagents.

Instrument Name	Purity	Manufacturer	Production Location
Sodium dodecylbenzene sulfonate (SDBS)	AR	Tianjin Bodi Chemical Industry Co., Ltd.	Tianjin, China
Polyisobutylene succinimide (T154)	AR	Tianjin Bodi Chemical Industry Co., Ltd.	Tianjin, China
Oleic acid diethanolamide (ODEA)	AR	Yousuo Chemical Industry	Linyi, Shandong, China
NaCl	CP	Sinopharm Chemical Reagent Co., Ltd.	Shanghai, China
Diesel	Industrial goods	China Petroleum & Chemical Corporation (Sinopec)	Xi’an, Shaanxi, China
Deionized water	-	Self-prepared in laboratory	Qingdao, Shandong, China

**Table 3 polymers-17-01543-t003:** Table of orthogonal experimental design.

Levels	Factors		
	ODEA/%	T154/%	SDBS/%
1	2	0.2	0.1
2	2.5	0.4	0.2
3	3	0.6	0.3
4	3.5	0.8	0.4

**Table 4 polymers-17-01543-t004:** Table of basic physical property parameters of cores.

Number	Porosity (*φ*)/%	Core Volume (*V*_f_)/mL	Core Length (*L*)/cm	Core Cross-Sectional Area (*A*)/cm^2^	Water-Measured Permeability (*k*)/10^−3^ μm^2^
1	19.75	24	4.920	4.88	2.37
2	22.95	21	5.186	4.05	7.76
3	20.53	40	8.094	4.94	8.01
4	21.31	29	6.950	4.17	12.34
5	22.41	32	6.586	4.86	13.96
6	22.08	31	6.602	4.70	16.06
7	23.97	32	6.570	4.87	19.96
8	25.30	27	5.602	4.82	20.39
9	26.45	27	5.520	4.86	22.16
10	25.85	33	6.834	4.83	29.48
11	29.80	25	5.366	4.66	29.80
12	39.45	29	6.646	4.36	31.73
13	27.96	25	5.418	4.61	32.62
14	33.66	35	7.138	4.90	34.85
15	35.67	24	5.142	4.67	36.72
16	35.78	31	6.598	4.70	37.33
17	36.54	34	6.985	4.87	39.31
18	40.21	30	6.254	4.80	51.59
19	45.68	32	6.542	4.89	58.37
20	47.00	28	6.630	4.22	65.41
21	46.97	29	5.324	4.38	69.90

**Table 5 polymers-17-01543-t005:** The liquid separation rate of the surfactant emulsion standing for 7 days at 25 °C.

Content/%	0.5	1	1.5	2	2.5	3	3.5	4	4.5
ODEA separation rate/%	1.20	1.03	1.12	0.98	1.01	0.43	0.67	0.98	0.65
T154 separation rate/%	15.35	14.41	12.04	13.04	13.21	14.03	11.95	12.47	13.02
SDBS separation rate/%	11.03	10.45	10.06	9.87	10.01	8.78	9.05	9.12	8.96

## Data Availability

The original contributions presented in this study are included in the article. Further inquiries can be directed to the corresponding author.

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
