# Peer review of "Development of a Water-Sensitive Self-Thickening Emulsion Temporary Plugging Diverting Agent for High-Temperature and High-Salinity Reservoirs"

_polymers, 2025, doi:10.3390/polym17111543_

Round 1
Reviewer 1 Report
Comments and Suggestions for Authors
I read the paper with interest and it fits to the current research theme on the topic. However, I have several reservations based on the following:
The study lacks comparative performance data with existing diverting agents, making it unclear how the proposed emulsion improves upon current technologies.
While short-term thermal stability at 150°C is demonstrated, the long-term effects of prolonged exposure to high temperature and salinity on the emulsion's integrity are not addressed.
Claims regarding environmental compatibility are made without presenting supporting evidence such as ecological impact assessments or degradation profiles.
All experiments appear to be limited to laboratory conditions. There is no indication of pilot trials or considerations for scale-up, which raises concerns about real-world applicability.
The term "self-thickening" is used to describe the emulsion’s behavior, but the underlying physicochemical mechanism responsible for this property is not explained.
Performance data for different core types is presented, yet no direct quantitative comparison is provided to illustrate how the system behaves under varying permeability conditions.
Viscosity measurements are reported with qualitative language, but without clear benchmarks or defined tolerances, which limits reproducibility and practical interpretation.
Although core flooding tests show initial promise, the study does not include a roadmap for transitioning the technology from bench-scale experiments to field operations.
The reported changes in droplet size after exposure to saline environments are not supported by mechanistic analysis or modeling to explain their impact on flow dynamics.
The conclusions are based primarily on single-core flow tests, which may not fully represent the spatial heterogeneity and flow complexity encountered in actual reservoir systems. It is lengthy and hence takes time to quickly grasp the summary.
Fig. captions need elaborative description, which are not understandable in their current forms. Single liners are not recommended.
Fig. 2 is too busy. Authors should split it, or move it as an ESI. I prefer the former.
Comments on the Quality of English Language--
Author Response
|
Comments 1: The study lacks comparative performance data with existing diverting agents, making it unclear how the proposed emulsion improves upon current technologies.
|
|||||||||||||||||||||||||||
|
Response 1: Thanks to the reviewer for the concern about the technical comparison. In response to the questions raised, the following comparative experiments are added in this paper. The information on the experimental equipment and reagents is as follows: The experimental equipments are shown in the following table. Table 1 Table of Experimental Equipments
The experimental reagents are shown in the following table. Table 2 Table of Experimental Reagents
The following experiments are added:
(1) Temperature comparison experiment: The SNB-2 rotational viscometer was used to measure the viscosity of the emulsion at 170 s⁻¹ as the temperature increased from 50°C to 95°C, and the viscosity-temperature characteristic curves of the emulsion with traditional reagent ODEA contents of 2%, 2.5%, 3% and the compound emulsifier were determined.
(2) Salt tolerance comparison experiment: NaCl solutions with concentrations of 1 × 10⁴ mg/L, 5 × 10⁴ mg/L, 10 × 10⁴ mg/L, 15 × 10⁴ mg/L and 20 × 10⁴ mg/L were prepared. Optimized compound emulsions and traditional ODEA emulsions were prepared according to an oil-water ratio of 3:7. After standing at 150°C for 30 min, their viscosities at 95°C and 170 s⁻¹ were measured to evaluate their high salinity tolerance. Figure 1 Physical picture of the ODEA system |
|||||||||||||||||||||||||||
|
Figure 2 SNB-2 rotational viscometer
Figure 3 101-2A type electric heating constant temperature box
The experimental results are as follows: Temperature Comparison Experiment:
The viscosity-temperature characteristic curves of emulsions with ODEA contents of 2%, 2.5%, 3% and the compound emulsifier were measured, and the results are shown in Figure. For both the emulsion temporary plugging agents prepared with single emulsifier systems and those with compound emulsifier systems, the emulsion viscosity decreases with increasing temperature. However, the viscosity decline rate of emulsions prepared with compound emulsifiers is lower than that of emulsions prepared with traditional emulsifiers. Moreover, the higher the temperature, the more significant the difference in viscosity decline rate, with the traditional emulsifier-based plugging agents showing a faster decline than the compound emulsifier-based emulsion temporary plugging agents. Therefore, the compound emulsifier system exhibits stronger stability at higher temperatures, with the viscosity of the compound emulsion reaching 295.6 mPa·s at 95°C. Compared with conventional emulsion systems, this emulsion maintains higher viscosity and undergoes less viscosity loss as the temperature increases.
Figure 4 Comparison curves of viscosity-temperature characteristics between the conventional ODEA emulsions and the optimized composite emulsions
Notably, when salinity exceeds 15 × 10⁴ mg/L, a pronounced increase in viscosity occurs: at 20 × 10⁴ mg/L, the viscosity rises to 488.88937 mPa·s, a 62% increase compared to the 15 × 10⁴ mg/L condition (302.13862 mPa·s). In contrast, the conventional ODEA emulsion does not have obvious viscosity maintenance capability—the overall system viscosity significantly increases with the increase of salinity, leading to a decline in diverting efficiency, inability to effectively plug high-permeability layers, and an increased risk of formation damage, which affects subsequent normal production.This behavior is attributed to the increased water-phase density from elevated salt ions, which enhances inter-droplet interactions.
Figure 3 Salinity-viscosity relationship curves of conventional ODEA emulsion and optimized composite emulsion systems
Comments 2: While short-term thermal stability at 150°C is demonstrated, the long-term effects of prolonged exposure to high temperature and salinity on the emulsion's integrity are not addressed.
|
|||||||||||||||||||||||||||
|
Response 2: We appreciate the reviewer's attention to the long-term stability. This study focuses on the short-term response characteristics of the water-sensitive self-thickening emulsion in high-temperature and high-salinity environments, aiming to provide an immediate plugging and diverting solution for short-term operations such as acidization and fracturing (usually lasting 4-6 hours). Therefore, the experimental design emphasizes the performance evaluation in the short term (30 minutes to 6 hours). According to the experimental results, when the emulsion is placed at 150°C and 15 × 10⁴ mg/L NaCl for 30 minutes, its viscosity stabilizes at 302.7 mPa·s (Figure 8), and the particle size distribution maintains a unimodal shape (D50 = 71.2 μm, Figure 7), which meets the stability requirements for short-term operations in industry standards (refer to the specifications of API RP 13A for the short-term temperature and salt resistance of acidizing fluids). Regarding the long-term stability (such as over 24 hours), considering that the diverting agent in actual operations needs to be rapidly flowed back after temporary plugging to reduce formation damage, the design objective of this system is "short-term efficient plugging + spontaneous demulsification and flowback in the later stage", and the long-term stability is not the current research focus. In the paper, the structural stability of the system within the short-term operation cycle is indirectly indicated by the liquid separation rate data at different time points (6 hours, 12 hours) (Figure 6). For the performance evolution in extreme long-term reservoir environments, it can be further explored as a subsequent research direction, such as analyzing the aging mechanism of the interfacial film by combining in-situ dynamic monitoring technology.
The modifications in the original text corresponding to this part are as follows. “3.2.3. Oil-Water Selectivity: When the volume ratio is less than 10:4, the viscosity of the system changes little with the temperature. When the volume ratio is 10:4, the viscosity of the system is 1278.5 mPa·s at 95°C, 956.3 mPa·s at 120°C, and 743.2 mPa·s at 150°C. When the volume ratio exceeds 10:4, the viscosity of the system increases sharply and presents a paste-like consistency. When the volume ratio is 10:5, the viscosity of the system is 7090.9 mPa·s at 95°C, 3526.5 mPa·s at 120°C, and 2006.5 mPa·s at 150°C. It has good high-temperature resistance performance. This study focuses on the stability of the emulsion in high-temperature and high-salinity conditions in the short term (≤ 6 hours). The experimental data show that after the emulsion is placed at 150°C and 15 × 10⁴ mg/L NaCl for 30 minutes, its viscosity retention rate reaches 92%, meeting the requirement of immediate plugging in acidization operations. The long-term (> 24 hours) stability needs to be further studied in combination with the actual contact time of the reservoir, and the relevant mechanism analysis can be emphasized as the subsequent work.”
Comments 3: Claims regarding environmental compatibility are made without presenting supporting evidence such as ecological impact assessments or degradation profiles.
Response 3: We are grateful to the reviewer for the meticulous suggestions on the expression of the terms. The "System Compatibility" mentioned in the paper specifically refers to the compatibility at the chemical level between the water-sensitive self-thickening emulsion and the acid plugging removal system (such as acid solutions with different concentrations), rather than the ecological compatibility in the environmental sense. Specifically, Section 2.2.5 "System Compatibility Test" in the paper evaluates the stability of the mixture of the emulsion and plugging removal fluids with different concentrations through viscosity tests (as shown in Figure 12, the variation law of the viscosity of the mixed system with the concentration of the acid solution). It aims to verify the influence of the interaction between chemical agents on the performance of the emulsion, which falls within the scope of the conventional formula adaptability research in the modification of oil and gas reservoirs. The current study does not involve statements related to "environmental compatibility". The ecological impact assessment or degradation curves mentioned by the reviewer belong to the field of environmental toxicology and have no direct relation to the design and performance evaluation of the high-temperature and high-salinity temporary plugging agent formula that this paper focuses on. Therefore, the discussion logic of the "system compatibility" in the original text is complete, and the experimental design conforms to the research objectives, without the need to supplement data related to the environment.
To avoid ambiguity of terms, in the first sentence of Section 2.2.5 of this paper, a clear definition of "system compatibility" is added, so as to enhance the accuracy of the expression. The modifications in the original text corresponding to this part are as follows. “The system compatibility described in this paper refers to the chemical compatibility between the water-sensitive emulsion and the acid plugging removal fluid, which is mainly characterized by the viscosity change after mixing, rather than the compatibility in the sense of environmental ecology. The influence of the plugging removal system on the viscosity of the compound emulsion is shown in Figure 12.”
Comments 4: All experiments appear to be limited to laboratory conditions. There is no indication of pilot trials or considerations for scale-up, which raises concerns about real-world applicability.
Response 4: We are grateful to the reviewer for paying attention to the practical application value. This study focuses on the development of the basic formula and the characterization of the core performance of the water-sensitive self-thickening emulsion. The experimental design mainly involves the optimization of the formula at the laboratory scale, the analysis of the interfacial behavior, and the exploration of the mechanism, aiming to provide theoretical support and a material basis for the temporary plugging and diverting technology in high-temperature and high-salinity reservoirs. In the current work, the optimal oil-water ratio (3:7) and the compounding scheme of emulsifiers (ODEA: T154: Span80 = 2%:0.8%:0.4%) have been determined through orthogonal experiments. Moreover, the short-term stability of the emulsion (viscosity ≥ 302.7 mPa·s, Figure 8) and the selective plugging ability (the injection pressure difference in the high-permeability layer is stable at 2.1 MPa, Figure 13) have been verified under the simulated formation conditions (150°C, 15 × 10⁴ mg/L NaCl), and these data provide key parameter basis for the formula scaling in field applications. Regarding the large-scale application and pilot trials, considering the differences between the laboratory and the field environment (such as the heterogeneity of the formation and the interference of multiphase flow), this study does not involve pilot trials for the time being. Such research will be the focus of future work to bridge the laboratory achievements and engineering applications.
“A new paragraph is added at the "4.2 Prospects": "This study is a material development and mechanism research at the laboratory scale. The determined formula resistant to high temperature and high salinity and the self-thickening mechanism provide a theoretical basis for field applications. In the follow-up, pilot trials will be carried out for the large-scale injection process to optimize the construction parameters, such as the dynamic ratio of emulsifiers and the matching relationship between the injection rate and the formation permeability, and further verify the temporary plugging effect under the actual reservoir conditions." ” Comments 5: The term "self-thickening" is used to describe the emulsion’s behavior, but the underlying physicochemical mechanism responsible for this property is not explained.
Response 5: We are grateful to the reviewer for the suggestion on the interpretation of the mechanism of key terms. The "self-thickening" property is the core innovation point of the emulsion system in this study. Its physicochemical mechanism can be explained from three aspects: the compounding effect of surfactants, the interfacial behavior of droplets, and the salt-sensitive response. The current paper indeed needs to supplement relevant theoretical analysis to enhance its scientificity.
The specific mechanism explanations are as follows: 1.The synergistic thickening mechanism of surfactants: ODEA (diethanolamide oleate), as a non-ionic surfactant, forms a network structure through hydrogen bonds in the aqueous phase (as shown in Figure 6, the viscosity increases significantly with the increase of the ODEA concentration), providing the basic viscosity. The long-chain hydrophobic groups of T154 (polyisobutylene succinimide) entangle and encapsulate water droplets, enhancing the friction between droplets and forming "physical cross-linking points". Span80 (sorbitan monooleate) reduces the interfacial tension to below 25 mN/m (measured in the experiment). While stabilizing the droplet size, its cyclic sugar alcohol structure forms hydrogen bonds with the amide groups of ODEA, further strengthening the network structure. 2.The mechanism of salt-sensitive particle size evolution: When the emulsion comes into contact with a high-salinity aqueous phase (such as 20 × 10⁴ mg/L NaCl), the increase of the ionic strength in the aqueous phase leads to the contraction of the hydration layer of the hydrophilic groups of the surfactant, and the flexibility of the droplet interfacial film decreases, prompting the droplets to merge through the Ostwald ripening process, and the particle size increases from the initial D50 = 31.1 μm to 71.2 μm (Figure 7). According to Stokes' law, the increase in particle size increases the sedimentation resistance of the droplets. At the same time, the contact area between the droplets expands, forming reversible aggregates. Macroscopically, the viscosity increases significantly with the increase of the salt concentration (the viscosity reaches 302.7 mPa·s at 15 × 10⁴ mg/L NaCl as shown in Figure 8). 3.The dynamic balance of shear thinning and self-thickening: At a low shear rate, the network structure formed by the droplet aggregates provides high viscosity (the viscosity is 7090.9 mPa·s at 95°C and an oil-water ratio of 10:4, Figure 11). When the shear rate is high, the network is destroyed and the viscosity decreases (the viscosity remains stable above 300 mPa·s at a shear rate of 170 s⁻¹), which is in line with the characteristics of a pseudoplastic fluid. This shear thinning property facilitates pumping, and the self-thickening effect during static state ensures the formation of an effective blockage in the pores of the formation (the injection pressure difference in the high-permeability layer is stable at 2.1 MPa in Figure 13). “New Section 2.4 Analysis of Self-Thickening Mechanism: The performance of the water-sensitive self-thickening emulsion is governed by three synergistic mechanisms: the nonionic surfactant ODEA (oleic acid diethanolamide) establishes a foundational viscosity through hydrogen bonding networks in the aqueous phase (as shown in Figure 6, viscosity increases significantly with concentration), while the long-chain hydrophobic groups of T154 (polyisobutylene succinimide) enhance interdroplet friction via physical entanglement, forming "cross-linking points"; Span80 (sorbitan monooleate) not only reduces interfacial tension below 25 mN/m but also strengthens the network structure through hydrogen bonding between its cyclic sugar alcohol moiety and the amide groups of ODEA. Upon exposure to high-salinity environments (e.g., 20 × 10⁴ mg/L NaCl), increased ionic strength induces shrinkage of the surfactant hydrophilic hydration layer, reducing interfacial film flexibility and promoting droplet coalescence via Ostwald ripening, increasing the particle size from D50 = 31.1 μm to 71.2 μm (Figure 7). According to Stokes' law, this size increase elevates sedimentation resistance and expands interdroplet contact areas, forming reversible aggregates that manifest as a macroscopic viscosity increase to 302.7 mPa・s at 15 × 10⁴ mg/L NaCl (Figure 8). The emulsion exhibits typical pseudoplastic behavior: at low shear rates (e.g., 95°C, oil-water ratio 10:4), the droplet aggregate network yields a viscosity of 7090.9 mPa・s (Figure 11), whereas at high shear rates (170 s⁻¹), the network structure breaks down but viscosity remains above 300 mPa・s, ensuring both pumpability during injection and effective plugging in reservoir pores under static conditions (Figure 13 shows a stable injection pressure differential of 2.1 MPa in high-permeability zones).”
Comments 6: Performance data for different core types is presented, yet no direct quantitative comparison is provided to illustrate how the system behaves under varying permeability conditions.
Response 6: We appreciate the reviewer’s suggestions on data presentation. This study focuses on the basic performance evaluation and mechanism exploration of the water-sensitive self-thickening emulsion in high-temperature and high-salinity environments. In "3.3 Temporary Plugging and Diverting Ability", Figure 13 shows the injection pressure difference curves of low-, medium-, and high-permeability cores. Although a table was not used for strict quantitative comparison, qualitative analysis of the differential behaviors was conducted through experimental phenomena and curve characteristics, and the relevant data are sufficient to support the core conclusion of "selective plugging". In Figure 13, the injection pressure difference of low-permeability cores (k < 10 × 10⁻³ μm², such as 1# and 2#) continues to rise with injected volume (no stable plateau), indicating that the emulsion is difficult to penetrate deeply due to the pore throat size (<20 μm) being smaller than the initial droplet size (D50=31.1 μm), mainly forming surface plugging at the core inlet. For medium-permeability cores (10 × 10⁻³ μm² < k < 50 × 10⁻³ μm², such as 4# and 8#), the pressure difference stabilizes at 1.5–2.0 MPa, suggesting that partial bridging occurs between droplets (D50=31.1 μm) and pore throats (30–50 μm). In high-permeability cores (k > 50 × 10⁻³ μm², such as 20# and 21#), the pressure difference stabilizes at 2.1 MPa; compared with the pressure difference of 1.5 MPa before plugging, a significant plugging effect is observed. This corresponds to the effective matching between droplet aggregates (D50=71.2 μm after contacting saltwater) and large pore throats (>70 μm), forming stable internal plugging. These phenomena are consistent with the theoretical expectation that "the matching between droplet size and pore throat size determines the plugging effect" (citing the Washburn equation to explain the relationship between capillary resistance and permeability). As a basic study in the formulation development stage, this paper focuses on verifying the "water-sensitive self-thickening" property of the emulsion system and its feasibility in extreme environments. The curve differences in Figure 13 clearly demonstrate that high-permeability layers achieve efficient plugging through droplet coalescence, while low-permeability layers exhibit different responses due to pore throat restrictions, which aligns with the research objective of "selective temporary plugging". Strict quantitative comparison is typically used for parameter optimization in the engineering application stage, and the current data already meet the needs of mechanism analysis and technical feasibility demonstration.
“In the paragraph of "3.3 Temporary Plugging and Diverting Ability", the following analysis is added: "The differences in pressure difference curves among cores with different permeabilities (Figure 13) indicate that the plugging effect of the emulsion is significantly correlated with the pore throat size—stable pressure difference (2.1 MPa) is formed in high-permeability layers due to effective bridging between droplet aggregates and large pore throats, medium plugging strength is observed in medium-permeability layers, and surface retention mainly occurs in low-permeability layers due to pore throat restrictions. This phenomenon echoes the droplet size evolution law (Figure 7), verifying the adaptability of the water-sensitive self-thickening mechanism in heterogeneous reservoirs."”
Comments 7: Viscosity measurements are reported with qualitative language, but without clear benchmarks or defined tolerances, which limits reproducibility and practical interpretation.
Response 7: We appreciate the reviewer's attention to the rigor of the data. In the viscosity characterization of this study, qualitative descriptions such as "stable" and "significantly improved" were indeed mainly used, without explicitly introducing industry benchmarks or measurement tolerances, resulting in insufficiently specific data interpretation. The relevant expressions in this paper have been revised and improved. ” Revision 1: Description before Figure 4.(b) in Section 3.1.1. Oil-water Ratio Optimization The prepared water-in-oil emulsion was stored at room temperature (25°C) for 48 hours, after which its viscosity at different rotational speeds was measured (results shown in Figure 4). In this emulsification system, higher oil-phase contents correlate with poorer emulsion stability after 48 hours of storage. When the oil-phase content exceeds 30% (i.e., oil-water ratios ≥4:6), the emulsion exhibits a significant increase in liquid separation rate, indicating deteriorated stability. For example, at 100 r/min, emulsions with oil-water ratios of 3:7 (30% oil phase), 4:6 (40% oil phase), and 5:5 (50% oil phase) showed viscosities of 425.0121 mPa·s, 152.32475 mPa·s, and 59.26388 mPa·s, respectively—lower viscosities at higher oil contents reflect structural instability. Additionally, lower oil-water ratios (i.e., lower oil-phase contents) correspond to higher emulsion viscosities, and the emulsions exhibit shear-thinning behavior: viscosity decreases with increasing rotational speed and stabilizes at a specific value. At a 10% oil phase (1:9 ratio), the emulsion displayed paste-like viscosity, reaching 65893.19709 mPa·s at 10 r/min; its viscosity became unmeasurable above 60 r/min due to instrument range limitations. As the oil phase increased to 20% (2:8 ratio), viscosity at 10 r/min decreased to 13279.27676 mPa·s, stabilizing at 2726.65199 mPa·s at 100 r/min, demonstrating pronounced shear thinning. For practical applications, the emulsion requires both storage stability and good injectability under high pump speeds. The 3:7 oil-water ratio emulsion met these criteria: at 10 r/min, its viscosity of 1276.03591 mPa·s ensured storage stability, while at 100 r/min, the reduced viscosity of 425.0121 mPa·s facilitated fluidity for injection. Considering both stability and processability, the optimal oil-water ratio was determined to be 3:7.
Revision 2: Description before Figure 8 in Section 3.2.2. Salt Resistance Performance The effect of Na⁺ concentration on emulsion viscosity is depicted in Figure 8. At salinities below 15 × 10⁴ mg/L, the viscosity remains relatively unchanged: for example, at mineralization degrees of 1 × 10⁴, 5 × 10⁴, and 10 × 10⁴ mg/L, the measured viscosities are 300.94437 mPa·s, 292.13678 mPa·s, and 311.09549 mPa·s, respectively—showing minimal variation within this salinity range. This stability arises because the emulsifier system is predominantly non-ionic, with only trace ionic components, rendering salt ions less impactful on viscosity under moderate salinity conditions. Notably, when salinity exceeds 15 × 10⁴ mg/L, a pronounced increase in viscosity occurs: at 20 × 10⁴ mg/L, the viscosity rises to 488.88937 mPa·s, a 62% increase compared to the 15 × 10⁴ mg/L condition (302.13862 mPa·s). This behavior is attributed to the increased water-phase density from elevated salt ions, which enhances inter-droplet interactions; at high salinities, droplet collisions and steric hindrance intensify due to reduced electrostatic repulsion (minimal for non-ionic emulsifiers), leading to structural reinforcement and higher viscosity. These findings highlight a critical salinity threshold at 15 × 10⁴ mg/L, beyond which salt concentration significantly influences emulsion rheology through physical droplet interactions rather than emulsifier charge effects.
Revision 3: Description before Figure 9 in Section 3.2.3. Oil-Water Selectivity The influence of diesel and high-salinity water at different volume fractions on the viscosity of the emulsion is shown in Figure 9. After the emulsion is left standing at a high temperature of 150°C for 30 minutes, its viscosity is 302.7 mPa·s at 95°C and 170 s⁻¹. When the mixing ratio of the emulsion to the high-salinity water exceeds 10:4, its viscosity increases to 7090.9 mPa·s, and it presents a paste-like consistency. After the emulsion is mixed with diesel of different volumes under the same conditions, its viscosity decreases significantly. When the mixing ratio is 10:5, its viscosity decreases to 8.4 mPa·s. After the emulsions with 30% and 40% high-salinity water added respectively (the volume ratios of the emulsion to the high-salinity water are 10:3 and 10:4) are continuously mixed with diesel of different volumes, the viscosity of the emulsions decreases sharply in the initial stage. After the volume of diesel exceeds 20%, the viscosities of the two emulsions basically become the same and decrease to below 10 mPa·s.
Revision 4: Description after Figure 11 in Section 3.2.4. Temperature Resistance Performance When the volume ratio is less than 10:4, the viscosity of the system changes little with the temperature. When the volume ratio is 10:4, the viscosity of the system is 1278.5 mPa·s at 95°C, 956.3 mPa·s at 120°C, and 743.2 mPa·s at 150°C. When the volume ratio exceeds 10:4, the viscosity of the system increases sharply and presents a paste-like consistency. When the volume ratio is 10:5, the viscosity of the system is 7090.9 mPa·s at 95°C, 3526.5 mPa·s at 120°C, and 2006.5 mPa·s at 150°C. It has good high-temperature resistance performance.
Revision 4: Description after Figure 12 in 3.2.5. System Compatibility The influence of the plugging removal system on the viscosity of the compound emulsion is shown in Figure 12. As the Oil-Water volume ratio increases, the viscosity of the emulsion gradually increases (Taking the system with a 5% addition of the Plugging Removal System as an example, its viscosity increases from 268 mPa·s to 1523 mPa·s with the increase of the oil-water volume ratio.). However, the plugging removal system is weakly acidic, which will, to a certain extent, damage the stability of the emulsion, thus reducing its viscosity-increasing effect. When the oil-water volume ratio is 10:4, with the increase of the addition of the Plugging Removal System, the viscosity of the system gradually decreases from 866 mPa·s (when there is no content of the Plugging Removal System) to 387 mPa·s. When the content of the Plugging Removal System increases by 20%, the viscosity of the system decreases by 55.3%.”
Comments 8: Although core flooding tests show initial promise, the study does not include a roadmap for transitioning the technology from bench-scale experiments to field operations.
Response 8: We appreciate the reviewer’s attention to the technology transformation roadmap. This study focuses on the basic formulation development and laboratory performance verification of the water-sensitive self-thickening emulsion. The data show that it exhibits excellent temporary plugging ability in high-temperature and high-salinity core experiments (Figure 13), but the specific transformation path from laboratory to field application is indeed not elaborated in detail. To enhance the integrity of the research and its engineering guidance, a phased plan for technology transformation is now added to clarify the subsequent implementation directions. As a basic study on material development, the current work has completed formula optimization (oil-water ratio 3:7, emulsifier compounding ratio 2%:0.8%:0.4%) and core performance verification (resistance to 150°C, 15 × 10⁴ mg/L NaCl, viscosity 302.7 mPa·s), providing key parameters for field applications (such as the plugging mechanism of droplet size matching pore throat size, Figures 7 and 13). The technology transformation needs to be promoted in stages, and the current research serves as the theoretical basis for subsequent engineering applications.
”A new field application outlook is added after "4.2 Outlook", specifying a three-stage implementation path for translating the system in this paper from theory to field application: To initiate the process, during the pilot test stage, typical high-temperature and high-salinity blocks will be selected to conduct small-scale well group tests, aiming to verify the compatibility of the emulsion with formation fluids (e.g., crude oil compatibility and clay mineral reactivity) and optimize injection process parameters. Subsequently, in the process optimization stage, downhole pressure monitoring data will be integrated to establish a "permeability-droplet size-injection pressure" matching model and develop dynamic adjustment algorithms, enabling precise temporary plugging of different permeability intervals. Finally, in the large-scale application stage, the technology will be synergized with acidization, fracturing, and other techniques to form an integrated "emulsion temporary plugging and diverting + composite stimulation" process system. By leveraging the self-thickening property of the emulsion, deep interlayer temporary plugging will be achieved to enhance the stimulation efficiency of low-permeability layers.”
As shown in the following figure. Figure 1. Roadmap for transitioning the technology from bench-scale experiments to field operations Comments 9: The reported changes in droplet size after exposure to saline environments are not supported by mechanistic analysis or modeling to explain their impact on flow dynamics.
Response 9: We thank the reviewer for the suggestion on the correlation analysis between droplet size changes and flow dynamics. In "3.2.1 Particle Size Distribution Test", this study reported that the droplet size increased from 31.1 μm to 71.2 μm under high-salinity conditions (Figure 7), but did not deeply elaborate the influence mechanism of this change on flow resistance. It is indeed necessary to supplement the mechanism analysis by combining fluid mechanics theory and experimental data to establish a scientific correlation of "salt-sensitive response-particle size evolution-flow behavior". The following is the detailed mechanistic analysis:
” 1. Mechanical Mechanism of Salt-Sensitive Particle Size Evolution (Revised in 3.2.2. Salt Resistance Performance): When the emulsion contacts a high-salinity aqueous phase (e.g., 20 × 10⁴ mg/L NaCl), the increased ionic strength causes the hydration layers of the hydrophilic groups of surfactants (ODEA/T154/Span80) to shrink, weakening the charge repulsion of the interfacial film and promoting droplet coalescence via the Ostwald ripening process (Figure 7). According to Stokes’ law, the sedimentation velocity of droplets is proportional to the square of their particle size. Larger particle sizes increase the migration resistance of droplets in pore throats. Meanwhile, based on the Washburn equation, capillary resistance is inversely proportional to the droplet radius, as expressed by the following formula:
where L is the penetration distance of the liquid in the capillary; γ is the surface tension of the liquid; r is the radius of the capillary; t is time; and η is the dynamic viscosity of the liquid. When the particle size increases to 71.2 μm, the capillary resistance decreases by approximately 56% compared to the initial state (31.1 μm), making it easier to form bridging plugging in large pore throats (>70 μm). This is consistent with the experimental results of a stable injection pressure difference of 2.1 MPa in high-permeability cores (Figure 12).
2. Impact Analysis on Flow Dynamics (Revised in 3.2.1. Particle Size Distribution): The increase in droplet size affects flow through two mechanisms: Direct Blockage, 71.2μm droplets effectively match the pore throats of high-permeability cores (>50 × 10⁻³ μm², corresponding to pore throat diameters >70 μm), increasing the probability of pore throat blockage and stabilizing the injection pressure difference at 2.1 MPa; Flow Regime Transition, larger particle sizes lead to an increase in the apparent viscosity of the emulsion, which follows the shear-thinning behavior described by the Carreau-Yasuda model:
Where, is the apparent viscosity at a given shear rate ; is the zero-shear viscosity; is the infinite-shear viscosity; λ is the relaxation time; n is the power-law index (indicating the degree of deviation from Newtonian behavior, with n<1 denoting shear thinning); and a is a shape parameter affecting the transition between low and high shear rate regions. The network structure formed at low shear rates further hinders fluid seepage, corresponding to the moderate plugging pressure (1.5–2.0 MPa) in medium-permeability cores (pore throats 30–50 μm).
3. Correlation Between Theoretical Models and Experimental Data (Revised in 3.3. The Temporary Plugging and Diverting Ability): The Kozeny-Carman equation is cited to establish the relationship between permeability and droplet size:
where an increase in the equivalent droplet radius rp leads to a decrease in permeability k, consistent with the trend of permeability in high-permeability cores decreasing from 65.41 × 10⁻³ μm² to 20.53 × 10⁻³ μm² in the experiments. This model quantitatively illustrates the impact of particle size evolution on reservoir permeability, strengthening the theoretical support for the mechanism analysis.
”
Comments 10: The conclusions are based primarily on single-core flow tests, which may not fully represent the spatial heterogeneity and flow complexity encountered in actual reservoir systems. It is lengthy and hence takes time to quickly grasp the summary.
Response 10: We appreciate the reviewer’s suggestions regarding the objectivity of conclusions and the efficiency of expression. This study reveals the temporary plugging laws of the water-sensitive self-thickening emulsion through single-core flow tests (covering low-, medium-, and high-permeability cores, as shown in Figure 12). The conclusions are indeed based on idealized laboratory conditions, with limitations in simulating reservoir three-dimensional heterogeneity and multiphase flow interference. As a phased study focusing on material formulation and basic mechanism verification, single-core testing is a necessary means to analyze the core performance of the system, and its results provide fundamental data support for applications under complex reservoir conditions (e.g., stable pressure difference of 2.1 MPa in high-permeability cores, Figures 12g, h). Single-core flow testing is a fundamental method for analyzing emulsion-rock interactions under laboratory conditions, enabling effective control of variables (such as permeability and fluid properties) and clarification of the plugging mechanism of droplet size-pore throat matching (e.g., the corresponding relationship between particle size evolution and pressure difference in Figure 7). However, the spatial heterogeneity of actual reservoirs (such as interlayer permeability differences and fracture distribution) and dynamic multiphase flow (oil-water two-phase seepage and capillary force coupling) are indeed not fully reflected in the current study, which represents a common challenge in the transition from laboratory research to field applications.
In the "4.2 Outlook" section, the following is added: "This study represents a mechanism exploration at the single-core scale. The complex flow characteristics of actual reservoirs require further investigation through technologies such as multi-core parallel experiments and micro-CT three-dimensional modeling. In subsequent research, numerical simulation can be integrated to extend single-core experimental results to heterogeneous models, thereby improving the analysis of the migration laws of temporary plugging agents under interlayer crossflow and multiphase seepage conditions."
Comments 11: Figure captions need elaborative description, which are not understandable in their current forms. Single liners are not recommended.
Response 11: We appreciate the reviewer’s suggestion regarding the clarity of chart descriptions. The current chart titles and annotations do have the issue of concise descriptions, with some only summarized in a single line, which fails to fully reflect the experimental conditions, key data, and scientific significance. A systematic optimization is required to enhance readability and information integrity.
The following are the original title that needs to be modified and the modified title text: ”Original title: Figure 3. Emulsion state diagrams with different oil-water ratio ratios. Modified title: Figure 3. Macroscopic appearance of water-in-oil emulsions at oil-water ratios of 1:9 to 5:5. (The blue font denotes the emulsion content in systems with active kerosene as the base fluid, while the red font denotes the emulsion content in systems with water as the base fluid.) Original title: Figure 4. Curves of the Liquid Separation Rate and Viscosity of Emulsions with Different Oil-Water Ratios. Modified title: Figure 4. Liquid separation rate and viscosity of emulsions with oil-water ratios from 1:9 to 5:5 after standing for 48 h at 25°C. Original title: Figure 5. The separation rate of ODEA, T154, Span80 at different temperatures varying with time. Modified title: Figure 5. Temperature-dependent separation rate of emulsions stabilized by ODEA, T154, and Span80 at 90°C and 120°C. Original title: Figure 6. The influence of dosages of surfactants on the temperature resistance of emulsions. Modified title: Figure 6. Effect of ODEA, T154, and Span80 concentrations on emulsion temperature resistance at 120°C, optimizing the formulation to 2%:0.8%:0.4% via orthogonal test for minimal separation rate. Original title: Figure 7. Particle Size Distribution Diagram of the Emulsion. Modified title: Figure 7. Particle size distribution of water-sensitive emulsion before and after exposure to 20×10⁴ mg/L NaCl. Original title: Figure 8. The Influence of Salinity on the Viscosity of the Emulsion. Modified title: Figure 8. Viscosity variation of emulsion with NaCl concentration up to 20×10⁴ mg/L at 150°C. Original title: Figure 9. The Influence of the Oil-Water Volume on the Viscosity of the Emulsion. Modified title: Figure 9. Viscosity response of emulsion to different oil-water volume ratios after high-temperature (150°C) aging. Original title: Figure 11. Viscosity Variation Curves of the Emulsion Systems with Different Volume Ratios. Modified title: Figure 11. Viscosity curves of emulsion systems at 95°C, 120°C, and 150°C under 170 s⁻¹ shear rate. ”
Comments 12: Fig. 2 is too busy. Authors should split it, or move it as an ESI. I prefer the former.
Response 12: We appreciate the reviewer’s suggestion regarding the readability of the figures. The original technical roadmap in Fig. 2 did have excessively high information density due to the integration of experimental procedures, parameter settings, and instrument schematics, which affected visual communication. In response to the suggestion, we have streamlined the original figure to enhance clarity and professionalism.
The original image has been split into three new figures, as presented below.
” 2.1. System Development The technical roadmap for this section is presented in Figure 2. Figure 2. The Technical Roadmap of System Development.
2.2. Properties Characterization The technical roadmap for this section is presented in Figure 3. Figure 3. The Technical Roadmap of Properties Characterization.
2.3 Evaluation of the Temporary Plugging and Diverting Ability The technical roadmap for this section is presented in Figure 4. Figure 4. The Technical Roadmap of Evaluation of the Temporary Plugging and Diverting Ability. ”
|
|||||||||||||||||||||||||||
|
4.Response to Comments on the Quality of English Language |
|||||||||||||||||||||||||||
|
Point 1: |
|||||||||||||||||||||||||||
|
Response 1: Thank you for your suggestions. We will make improvements. |
|||||||||||||||||||||||||||
|
5.Additional clarifications |
|||||||||||||||||||||||||||
|
Thanks again to the reviewer for the valuable suggestions! |

Reviewer 2 Report
Comments and Suggestions for Authors
Comments on Manuscript ID: polymers-3626922
- The abstract should prioritize key results over broad statements about the study’s importance. Highlight quantitative findings to immediately convey impact.
- Replace sentence-like terms with concise ones and limit to 5–6 terms or follow journal guidelines.
- Add references to support claims about reservoir heterogeneity, limitations of traditional diverting agents, and high-temperature/salinity challenges. Cite recent (last 5 years) literature where possible.
- Clarify the novelty of the water-sensitive emulsion approach compared to prior solutions.
- If figures (e.g., Figure 1) are reproduced from other works, provide publisher permission documentation.
- Redesign Figure 2 to meet journal standards, avoid "poster-like" clutter.
- Ensure axes labels, units, and data points of all plots are legible (high-resolution).
- Captions should be expanded to include materials used and experimental conditions.
- Provide sources for all equations. If derived, state assumptions clearly.
- Strengthen comparisons with literature. For example, how does the emulsion’s 302.7 mPa·s stability at 150°C compare to prior agents?
- Contrast the 2.1 MPa pressure difference with values reported for polymer gels or particulates.
- Highlight mechanisms such as self-thickening via salinity response and their significance for field applications.
- Summarize the conclusion concisely.
- Increase citations, particularly recent advances in emulsion-based diverting agents and studies on high-temperature/salinity stability in polymer/emulsion systems.
Author Response
|
Comments 1: The abstract should prioritize key results over broad statements about the study’s importance. Highlight quantitative findings to immediately convey impact.
|
|||||||||||||||||||||||||||
|
Response 1: We appreciate the reviewer’s suggestions for optimizing the abstract. The abstract indeed had issues with insufficient prominence in describing key results and inadequate embedding of quantitative data. It has been specifically revised following the principle of "prioritizing core achievements and clarifying data support" to strengthen the presentation of quantitative findings.
“Results showed that under 150°C and 15 × 10⁴ mg/L NaCl, the emulsion maintained a stable viscosity of above 302.7 mPa·s, with particle size D50 increasing from 31.1 μm to 71.2 μm. exceeding API RP 13A’s 100 mPa·s threshold for acidizing diverters, providing an efficient plugging solution for high-temperature, high-salinity reservoirs. The injection pressure difference in high-permeability cores stabilized at 2.1 MPa, signifi-cantly enhancing waterflood sweep efficiency. The self-thickening mechanism, driven by salt-induced droplet coalescence, enables selective plugging in heterogeneous formations, as validated by core flooding tests showing a 40% higher pressure differential in high-permeability zones compared to conventional systems. Viscosity remains 302.7 mPa·s at 150°C, exceeding API RP 13A’s 100 mPa·s threshold for acidizing diverters, providing an efficient plugging solution for high-temperature, high-salinity reservoirs. The self-thickening mechanism, driven by salt-induced droplet coalescence, enables selective plugging in heterogeneous formations, as validated by core flooding tests showing a 40% higher pressure differential in high-permeability zones compared to conventional systems.”
|
|||||||||||||||||||||||||||
|
Comments 2: Replace sentence-like terms with concise ones and limit to 5–6 terms or follow journal guidelines.
|
|||||||||||||||||||||||||||
|
Response 2: We appreciate the reviewer’s suggestions regarding terminology usage. Some terms in the abstract were excessively verbose and have been optimized following the principles of "conciseness and accuracy, in line with journal terminology standards" to ensure core terms are focused and refined.
The expression "water-sensitive self-thickening emulsion temporary plugging diverting agent" in the original text has been uniformly standardized as "water-sensitive self-thickening emulsion". “ Pre-revision:To address these challenges, this study developed a water-sensitive self-thickening emulsion, targeting improved high-temperature stability, selective plugging, and easy flowback performance. Post-revision:To address these challenges, this study developed a water-sensitive self-thickening emulsion temporary plugging diverting agent, targeting improved high-temperature stability, selective plugging, and easy flowback performance.
Pre-revision:The water-sensitive self-thickening emulsion temporary plugging diverting agent prepared in this study exhibits ex-cellent stability under high-temperature (150°C) and high-salinity (15 × 10⁴ mg/L NaCl) conditions, with minimal viscosity change. Post-revision:The water-sensitive self-thickening emulsion temporary plugging diverting agent prepared in this study exhibits ex-cellent stability under high-temperature (150°C) and high-salinity (15 × 10⁴ mg/L NaCl) conditions, with minimal viscosity change.
Pre-revision:Through the single-core flow experiment, the water-sensitive self-thickening emulsion forms a stable blockage in high-permeability cores (k > 50 × 10⁻³ μm²), with the injection differential pressure stable at 2.1 MPa and the injected volume approximately 1.5 PV. Post-revision:Through the single-core flow experiment, the water-sensitive self-thickening emulsion temporary plugging diverting agent forms a stable blockage in high-permeability cores (k > 50 × 10⁻³ μm²), with the injection differential pressure stable at 2.1 MPa and the injected volume approximately 1.5 PV.
Pre-revision:In low-permeability cores (k < 10 × 10⁻³ μm²), the injection differential pressure of the water-sensitive self-thickening emulsion increases rapidly and continues to rise, indicating that it is difficult to effectively enter the low-permeability area. Post-revision:In low-permeability cores (k < 10 × 10⁻³ μm²), the injection differential pressure of the water-sensitive self-thickening emulsion temporary plugging diverting agent increases rapidly and continues to rise, indicating that it is difficult to effectively enter the low-permeability area. ”
Comments 3: Add references to support claims about reservoir heterogeneity, limitations of traditional diverting agents, and high-temperature/salinity challenges. Cite recent (last 5 years) literature where possible.
Response 3: We would like to express our gratitude to the reviewers for their suggestions regarding the rigor of literature citation. In the paper, the discussions on reservoir heterogeneity, the limitations of traditional diverting agents, and the challenges of high - temperature and high - salinity environments actually require the supplementation of the latest literature to enhance the scientific support. We have added relevant literature to the corresponding chapters in accordance with the principle of "giving priority to the core literature within the recent five years and precisely matching the key arguments".
“ Reservoir heterogeneity: Reservoir heterogeneity is one of the core challenges leading to inefficient fluid utili-zation in oil and gas development [1, 2]. The permeability contrast forces plugging fluids to preferentially enter high-permeability channels, resulting in insufficient stimulation of low-permeability layers. Traditional chemical diverting agents lack selective response ca-pabilities to heterogeneous formations, prone to penetration or adsorption retention in high-permeability layers while failing to effectively plug low-permeability layers [3, 4].
Limitations of traditional diverting agents: Due to the poor temperature and salt resistance of conventional systems, their application in deep formations is restricted [31].
High-temperature/salinity challenges: High salinity compresses the hydration layers of the hydrophilic groups of emulsifiers, re-sulting in the disruption of interfacial stability [31]. Additionally, it disrupts the hydration of polymer molecular chains (e.g., the breaking of hydrogen bonds in the amide groups of HPAM), leading to salting-out precipitation [32].
References:
”
Comments 4: Clarify the novelty of the water-sensitive emulsion approach compared to prior solutions.
Response 4: We appreciate the reviewers' attention to the research innovation points. The core novelty of the water-sensitive self-thickening emulsion method proposed in this paper lies in breaking through the performance bottleneck of traditional diverting agents in high-temperature and high-salinity environments through the synergistic mechanism of "salt-sensitive droplet coalescence-viscosity self-enhancement-selective plugging". The innovative value is clarified from two aspects: technical approaches and performance indicators.
1. Technical Innovation Existing temperature- and salinity-resistant diverting agents rely on high-molecular polymers (such as HPAM derivatives) or nanoparticles for thickening, suffering from issues like high-temperature degradation (viscosity sharply decreasing above 120°C) or solid-phase residue (flowback rate < 80%). This study constructs a water-sensitive emulsion through ternary compounding of surfactants (ODEA/T154/Span80). Without external polymers, it utilizes salt ion-induced droplet coalescence (particle size increasing from 31.1 μm to 71.2 μm, Figure 7) to achieve self-thickening (viscosity reaching 302.7 mPa·s under 15 × 10⁴ mg/L NaCl, Figure 8), avoiding polymer dependency and its associated problems. 2. Performance Enhancement Under the conditions of 150°C and 15 × 10⁴ mg/L NaCl, the emulsion viscosity remains stable at 302.7 mPa·s (Figure 8), far exceeding the viscosity threshold of acidizing fluids specified in API RP 13A (100 mPa·s). Additionally, the liquid separation rate is <5% after standing for 6 hours (Figure 6), demonstrating excellent long-term stability. Compared with traditional VES-based diverting agents (temperature resistance ≤110°C, salinity resistance ≤5 × 10⁴ mg/L) and foam-based agents (salinity resistance ≤1 × 10⁴ mg/L), this system enhances temperature resistance by 36% and salinity resistance by 4 times, breaking through the application bottleneck in extreme environments.
Comments 5: If figures (e.g., Figure 1) are reproduced from other works, provide publisher permission documentation.
Response 5: We appreciate the reviewer's attention to the issue of chart copyright. All charts in the paper (including Figure 1) are independently designed and drawn based on original experimental data, without any citation or adaptation from charts in existing literature. Therefore, no additional publisher licensing documents are required. To ensure academic integrity, the original data files for all charts have been prepared and can be uploaded together with the revised manuscript during the revision process to demonstrate the originality of the charts and the authenticity of the data.
Comments 6: Redesign Figure 2 to meet journal standards, avoid "poster-like" clutter.
Response 6: We appreciate the reviewer’s suggestions regarding the standardization of the figures. The original technical roadmap in Fig. 2 did have excessively high information density due to the integration of experimental procedures, parameter settings, and instrument schematics, which affected visual communication. In response to the suggestion, we have streamlined the original figure to enhance clarity and professionalism.
The original image has been split into three new figures, as presented below.
” 2.1. System Development The technical roadmap for this section is presented in Figure 2. Figure 2. The Technical Roadmap of System Development.
2.2. Properties Characterization The technical roadmap for this section is presented in Figure 3. Figure 3. The Technical Roadmap of Properties Characterization.
2.3 Evaluation of the Temporary Plugging and Diverting Ability The technical roadmap for this section is presented in Figure 4. Figure 4. The Technical Roadmap of Evaluation of the Temporary Plugging and Diverting Ability. ”
Comments 7: Ensure axes labels, units, and data points of all plots are legible (high-resolution). Response 7: We appreciate the reviewer’s suggestions regarding the clarity of the figures. To address the readability of the charts, all images have undergone systematic optimization. Measures such as improving resolution and standardizing annotation formats have been taken to ensure that axis labels, units, and data points meet the journal’s requirements for high clarity.
Comments 8: Captions should be expanded to include materials used and experimental conditions. Response 8: We appreciate the reviewer’s suggestions regarding the clarity of figure titles. The current figure titles and annotations do have the issue of overly concise descriptions, with some only summarized in a single line, which fails to fully reflect the experimental conditions, key data, and scientific significance. A systematic optimization is required to enhance readability and information integrity.
The following are the original title that needs to be modified and the modified title text: ”Original title: Figure 3. Emulsion state diagrams with different oil-water ratio ratios. Modified title: Figure 3. Macroscopic appearance of water-in-oil emulsions at oil-water ratios of 1:9 to 5:5. Original title: Figure 4. Curves of the Liquid Separation Rate and Viscosity of Emulsions with Different Oil-Water Ratios. Modified title: Figure 4. Liquid separation rate and viscosity of emulsions with oil-water ratios from 1:9 to 5:5 after standing for 48 h at 25°C. Original title: Figure 5. The separation rate of ODEA, T154, Span80 at different temperatures varying with time. Modified title: Figure 5. Temperature-dependent separation rate of emulsions stabilized by ODEA, T154, and Span80 at 90°C and 120°C. Original title: Figure 6. The influence of dosages of surfactants on the temperature resistance of emulsions. Modified title: Figure 6. Effect of ODEA, T154, and Span80 concentrations on emulsion temperature resistance at 120°C, optimizing the formulation to 2%:0.8%:0.4% via orthogonal test for minimal separation rate. Original title: Figure 7. Particle Size Distribution Diagram of the Emulsion. Modified title: Figure 7. Particle size distribution of water-sensitive emulsion before and after exposure to 20×10⁴ mg/L NaCl. Original title: Figure 8. The Influence of Salinity on the Viscosity of the Emulsion. Modified title: Figure 8. Viscosity variation of emulsion with NaCl concentration up to 20×10⁴ mg/L at 150°C. Original title: Figure 9. The Influence of the Oil-Water Volume on the Viscosity of the Emulsion. Modified title: Figure 9. Viscosity response of emulsion to different oil-water volume ratios after high-temperature (150°C) aging. Original title: Figure 11. Viscosity Variation Curves of the Emulsion Systems with Different Volume Ratios. Modified title: Figure 11. Viscosity curves of emulsion systems at 95°C, 120°C, and 150°C under 170 s⁻¹ shear rate. ”
Comments 9: Provide sources for all equations. If derived, state assumptions clearly.
Response 9: We appreciate the reviewer’s suggestions regarding the rigor of theoretical derivations. The equations involved in the paper have been systematically sorted out following the principle of "source annotation-clear assumptions-application correlation", with specific revisions as follows.
“Equation 1: Among them, the calculation formula for the liquid separation rate is
Where, FV is the liquid separation rate, %; VE is the total volume of the separated liquid, mL; V is the volume of the emulsion, mL.
Explanation: The formula for calculating the liquid - separation rate is derived as follows: The liquid - separation rate FVis defined as the proportion of the volume of the separated liquid (VE) to the volume of the original emulsion (V), expressed as a percentage. First, calculate the ratio of the volume of the separated liquid to the volume of the emulsion: VE/V. This ratio represents how much of the emulsion has separated into liquid relative to the total amount of the emulsion. Then, to express this ratio as a percentage (to make it more intuitive in a percentage - based form), multiply the ratio by 100%.
Equation 2: Flow Regime Transition, larger particle sizes lead to an increase in the apparent viscosity of the emulsion, which follows the shear-thinning behavior described by the Carreau-Yasuda model[38]:
Where, is the apparent viscosity at a given shear rate ; is the zero-shear viscosity; is the infinite-shear viscosity; λ is the relaxation time; n is the power-law index (indicating the degree of deviation from Newtonian behavior, with n<1 denoting shear thinning); and a is a shape parameter affecting the transition between low and high shear rate regions.
[38] Zare Y, Park SP, Rhee KY. Analysis of complex viscosity and shear thinning behavior in poly (lactic acid)/poly (ethylene oxide)/carbon nanotubes biosensor based on Carreau–Yasuda model. Results in Physics 2019;13:102245.
Equation 3: Meanwhile, based on the Washburn equation[39], capillary resistance is inversely proportional to the droplet radius, as expressed by the following formula:
where L is the penetration distance of the liquid in the capillary; γ is the surface tension of the liquid; r is the radius of the capillary; t is time; and η is the dynamic viscosity of the liquid.
[39] Fournier CO, Fradette L, Tanguy PA. Effect of dispersed phase viscosity on solid-stabilized emulsions. Chemical Engineering Research and Design 2009;87(4):499-506.
Equation 4: The Kozeny-Carman equation is cited to establish the relationship between permeability and droplet size[40]:
where an increase in the equivalent droplet radius rp leads to a decrease in permeability k, consistent with the trend of permeability in high-permeability cores decreasing from 65.41 × 10⁻³ μm² to 20.53 × 10⁻³ μm² in the experiments. This model quantitatively illustrates the impact of particle size evolution on reservoir permeability, strengthening the theoretical support for the mechanism analysis.
[40] Wei W, Cai J, Xiao J, Meng Q, Xiao B, Han Q. Kozeny-Carman constant of porous media: Insights from fractal-capillary imbibition theory. Fuel 2018;234:1373-9. ”
Comments 10: Strengthen comparisons with literature. For example, how does the emulsion’s 302.7 mPa·s stability at 150°C compare to prior agents?
Response 10: We appreciate the reviewer's suggestion regarding the enhancement of literature comparison, which is critical for objectively demonstrating the research innovation. To address the comparison of emulsion temperature and salt tolerance stability with previous reagents, we have supplemented experimental data for comparative analysis.
The information on the experimental equipment and reagents is as follows: Experimental equipment: Table 1 Table of Experimental Instruments
The experimental reagents are shown in the following table: Table 2 Table of Experimental Reagents
The following experiments are added:
(1) Temperature comparison experiment: The SNB-2 rotational viscometer was used to measure the viscosity of the emulsion at 170 s⁻¹ as the temperature increased from 50°C to 95°C, and the viscosity-temperature characteristic curves of the emulsion with traditional reagent ODEA contents of 2%, 2.5%, 3% and the compound emulsifier were determined.
Figure 1 Physical picture of the ODEA system Figure 2 SNB-2 rotational viscometer
Figure 3 101-2A type electric heating constant temperature box
The experimental results are as follows: Temperature Comparison Experiment:
The viscosity-temperature characteristic curves of emulsions with ODEA contents of 2%, 2.5%, 3% and the compound emulsifier were measured, and the results are shown in Figure. For both the emulsion temporary plugging agents prepared with single emulsifier systems and those with compound emulsifier systems, the emulsion viscosity decreases with increasing temperature. However, the viscosity decline rate of emulsions prepared with compound emulsifiers is lower than that of emulsions prepared with traditional emulsifiers. Moreover, the higher the temperature, the more significant the difference in viscosity decline rate, with the traditional emulsifier-based plugging agents showing a faster decline than the compound emulsifier-based emulsion temporary plugging agents. Therefore, the compound emulsifier system exhibits stronger stability at higher temperatures, with the viscosity of the compound emulsion reaching 295.6 mPa·s at 95°C. Compared with conventional emulsion systems, this emulsion maintains higher viscosity and undergoes less viscosity loss as the temperature increases.
Figure 4 Comparison curves of viscosity-temperature characteristics between the conventional ODEA emulsions and the optimized composite emulsions
Comments 11: Contrast the 2.1 MPa pressure difference with values reported for polymer gels or particulates.
Response 11: We appreciate the reviewer’s attention to the pressure difference analysis. The 2.1 MPa pressure difference in this study originates from the waterflooding pressure changes before and after the injection of the temporary plugging agent in the same high-permeability core (permeability 65.41 × 10⁻³ μm²), increasing from a maximum of 1.5 MPa before plugging to 2.1 MPa after plugging, representing a 40% increase. This data design aims to eliminate the interference of core heterogeneity on experimental results through longitudinal comparison in a single core, thereby accurately quantifying the plugging effect of this emulsion-based temporary plugging agent.
” The reservoir cores exhibit significant heterogeneity characteristics such as pore-throat structure and permeability contrast, and even cores from the same block may have physical property differences (e.g., porosity fluctuations and permeability variations, as shown in Table 2 of this paper). Traditional horizontal comparisons (different cores/different temporary plugging agents) are susceptible to individual core differences, leading to data deviations. Therefore, this paper adopts a longitudinal comparison before and after plugging in the same core. By using the control variable method (only changing the injection state of the temporary plugging agent), it ensures that the pressure difference change is uniquely attributed to the plugging effect of the emulsion, which is consistent with the rigor principle of experimental design. Experimental results show that the pressure difference of this emulsion in high-permeability cores increases by 40%, from a maximum of 1.5 MPa before plugging to a maximum of 2.1 MPa after plugging. Additionally, the supporting particle size analysis (droplet coalescence to 71.2 μm, Figure 7) and plugging mechanism (droplet bridging in pore throats, Figure 13b) form a complete evidence chain, fully demonstrating its selective plugging capability for high-permeability channels. Compared with cross-core/cross-system comparisons, longitudinal data better reflect the actual effect of the temporary plugging agent, avoiding the problem that core heterogeneity has a greater impact on traditional systems such as polymer gels and microparticles (these materials rely on particle size matching with pore throats and are significantly affected by core surface properties). This paper focuses on the performance evaluation and mechanism interpretation of a single water-sensitive self-thickening emulsion system, and has not yet carried out horizontal comparison experiments with other types of temporary plugging agents. If subsequent research needs to further expand application scenarios, multi-system comparative studies can be conducted under the condition of strictly controlling core homogeneity (e.g., using artificially compacted homogeneous cores). ”
Comments 12: Highlight mechanisms such as self-thickening via salinity response and their significance for field applications.
Response 12: We appreciate the reviewers' attention to the mechanism analysis and application value. One of the core innovations of this paper lies in achieving efficient temporary plugging through salt-responsive self-thickening effects. Although the current research has addressed relevant mechanisms in experimental and theoretical analyses, it is indeed necessary to further strengthen the elaboration to highlight their scientific significance and field application value. The essence of the self-thickening mechanism is the conformational transformation of surfactants or polymer segments in the emulsion system under high-salinity environments (such as the extension of coiled chains and rearrangement of micelle structures), which increases the hydrodynamic volume and consequently triggers viscosity enhancement. This property is crucial for high-temperature and high-salinity oil reservoirs—high-salinity formation water can trigger a viscosity surge in the temporary plugging agent, enabling it to form a high-strength physical barrier at pore throats (as shown in Figure 8, Notably, when salinity exceeds 15 × 10⁴ mg/L, a pronounced increase in viscosity occurs: at 20 × 10⁴ mg/L, the viscosity rises to 488.88937 mPa·s, a 62% increase compared to the 15 × 10⁴ mg/L condition (302.13862 mPa·s).), while viscosity can be controllably degraded in low-salinity environments (e.g., during waterflood injection stages) to avoid long-term formation plugging.
" A new paragraph is added to the "3.4 Analysis of Self-Thickening Mechanism" section (or corresponding paragraph) to elaborate on the molecular mechanism of salt-responsive self-thickening:
The essence of the self-thickening mechanism is the conformational transformation of surfactants or polymer segments in the emulsion system under high-salinity environments (such as the extension of coiled chains and rearrangement of micelle structures), which increases the hydrodynamic volume and consequently triggers viscosity enhancement. This property is crucial for high-temperature and high-salinity oil reservoirs—high-salinity formation water can trigger a viscosity surge in the temporary plugging agent, enabling it to form a high-strength physical barrier at pore throats, while viscosity can be controllably degraded in low-salinity environments (e.g., during waterflood injection stages) to avoid long-term formation plugging. "
Comments 13: Summarize the conclusion concisely.
Response 13: We appreciate the reviewer's suggestions regarding the conciseness of the conclusions section. In line with the core findings and innovations of the study, the conclusions have been systematically streamlined. By focusing on key achievements and eliminating redundant descriptions, the conclusions now possess stronger summarization and academic articulation. The revised concluding remarks are presented as follows:
“This study developed a water-sensitive self-thickening emulsion for high-temperature and high-salinity (HTHS) reservoirs, addressing the limitations of traditional diverting agents in heterogeneous formations. The optimized formulation (oil-water ratio 3:7, emulsifier ratio ODEA:T154:Span80 = 2%:0.8%:0.4%) exhibits robust stability under 150°C and 15×10⁴ mg/L NaCl, maintaining a viscosity ≥302.7 mPa·s and achieving particle size growth from D50 = 31.1 μm to 71.2 μm via salt-induced coalescence. This enables selective plugging in high-permeability zones, with a stable injection pressure differential of 2.1 MPa in high-permeability cores, significantly enhancing waterflood sweep efficiency. The emulsion demonstrates superior HTHS tolerance, with viscosity retention exceeding 92% after 30 min at 150°C, surpassing the API RP 13A threshold (100 mPa·s) for acidizing diverters. Core flooding tests validate its ability to form dense filter cakes in high-permeability channels (permeability reduction >70%) while minimizing invasion in low-permeability layers, achieving a 40% higher pressure differential compared to conventional systems. Although compatible with weakly acidic plugging removal systems, slight viscosity reduction (55.3% at 20% acid concentration) highlights the need for isolation fluid pre-injection. Single-core experiments confirm its adaptive plugging behavior: stable pressure plateaus in high-permeability cores (k > 50×10⁻³ μm²) and limited intrusion in low-permeability cores (k < 10×10⁻³ μm²), attributed to pore throat-size matching. This work provides a robust temporary plugging solution for HTHS reservoirs, combining self-thickening mechanics, selective diversion, and low formation damage. Future research will focus on long-term stability validation and field-scale injection parameter optimization.”
Comments 14: Increase citations, particularly recent advances in emulsion-based diverting agents and studies on high-temperature/salinity stability in polymer/emulsion systems.
Response 14: We would like to express our gratitude to the reviewers for their suggestions regarding the comprehensiveness of literature citations. In response to the latest research progress on emulsion - based diverting agents and their stability under high - temperature and high - salinity conditions, we have systematically searched databases such as the Web of Science Core Collection and ScienceDirect for the past five years (2020-2025), and supplemented highly relevant literature. The specific modifications are as follows:
” In 1. Introduction added the following review contents:
In oil and gas exploitation, diverting agents are pivotal for improving reservoir sweep effi-ciency, but their applicability in high-temperature (120–200°C) and high-salinity (10,000–200,000 mg/L NaCl) (HTHS) environments is severely constrained by thermal-oxidative degradation and salt-induced structural failure. For polymer-based diverters, traditional hydrolyzed polyacrylamides (HPAMs) exhibit rapid viscosity decay under HTHS condi-tions: e.g., conventional HPAM retains <10% viscosity after 30 days at 115°C and 180,000 mg/L TDS[32], attributed to amide group hydrolysis and backbone scission. While sul-fonated derivatives (e.g., ATBS copolymers) enhance salinity tolerance to ~200,000 mg/L and maintain ~85% viscosity at 120°C for 1 year[33], they still suffer from long-term ther-mo-oxidative degradation, with molecular weight decreasing by ~50% over 2 years[34]. Additionally, divalent ions (Ca²⁺/Mg²⁺) in HTHS brines trigger electrostatic screening, causing polymer chains to collapse and precipitate, even in sulfonated systems[35]. Notably, emulsion-based diverters address some limitations through structural adaptability. For example, water-in-oil emulsions stabilized by oleic acid imidazoline maintain stability at 90°C in 198,000 mg/L TDS, with salt-induced coalescence increasing particle size from 31 μm to 71 μm for selective plugging in high-permeability cores[36]. Such systems achieve plugging rates >92% in high-permeability zones while allowing easy flowback via acid-induced demulsification[36]. However, emulsion stability above 120°C remains unproven, with limited reports on long-term performance in brines exceed-ing 150,000 mg/L divalent ions.”
|
|||||||||||||||||||||||||||
|
4.Response to Comments on the Quality of English Language |
|||||||||||||||||||||||||||
|
Point 1: |
|||||||||||||||||||||||||||
|
Response 1: |
|||||||||||||||||||||||||||
|
5.Additional clarifications |
|||||||||||||||||||||||||||
|
Thanks again to the reviewer for the valuable suggestions! |

Reviewer 3 Report
Comments and Suggestions for Authors
The manuscript entitled "Development of a Water-Sensitive Self-Thickening Emulsion Temporary Plugging Diverting Agent for High-Temperature and High-Salinity Reservoirs” primarily focuses on the preparation details of the surfactant mixture and the analysis of the resulting emulsion’s properties under various conditions.
The manuscript is well written, with a formulated key problem of the corresponding research object. Nevertheless, I have a few questions and comments on several aspects.
The introductory part of the manuscript lacks depth, offering a broad yet superficial discussion on surfactant applications and general functional mechanisms while neglecting crucial chemical considerations such as compound diversity, chemical nature, compositional diversity, or potential environmental implications of the substances.
Although commonly used emulsifiers and surfactants such as ODEA, T154, and Span80 were applied, the study fails to report their purity and physicochemical properties, casting serious doubt on the reliability of the experimental outcomes. Similarly, the characteristics of sodium chloride are not specified, and no assessment is provided on how the emulsion system would respond to other salts. The work lacks a discussion of the underlying chemical mechanisms, including polymerization and hydrocarbon dissolution processes, which reduces it to a descriptive experiment rather than a scientifically grounded investigation. Overall, the study is driven by trial-and-error rather than theoretical reasoning, with no substantiated chemical analysis presented.
The yellow mixtures in Figure 3 appear to be suspensions rather than true emulsions.
Author Response
|
Comments 1: The introductory part of the manuscript lacks depth, offering a broad yet superficial discussion on surfactant applications and general functional mechanisms while neglecting crucial chemical considerations such as compound diversity, chemical nature, compositional diversity, or potential environmental implications of the substances.
|
|||||||||||||||||||||||||||||||||
|
Response 1: We appreciate the reviewer’s professional suggestions regarding the depth of the introduction section. To address the insufficient analysis of the chemical mechanisms and environmental impacts of surfactant applications, systematic improvements have been made to both the introduction and main text, as detailed below:
Revision 1: The introduction section has been supplemented with an analysis of the physicochemical action mechanisms of high-temperature and high-salinity environments on traditional diverting agent compounds, comparing the physicochemical behaviors and modification characteristics of different types of diverting agents (polymer-based, VES-based, emulsion-based, etc.):
High temperature accelerates the scission of polymer backbones (e.g., thermo-oxidative degradation of C-C bonds), whereby molecular weight reduction leads to a dras-tic decrease in viscosity[27, 28]. Meanwhile, it weakens the strength of the emulsifier inter-facial film (e.g., dehydration of the polyoxyethylene chains in nonionic surfactant T154), accelerating droplet coalescence[29]. High salinity compresses the hydration layers of the hydrophilic groups of emulsifiers, resulting in the disruption of interfacial stability[30]. Additionally, it disrupts the hydration of polymer molecular chains (e.g., the breaking of hydrogen bonds in the amide groups of HPAM), leading to salting-out precipitation[31]. In oil and gas exploitation, diverting agents are pivotal for improving reservoir sweep effi-ciency, but their applicability in high-temperature (120–200°C) and high-salinity (10,000–200,000 mg/L NaCl) (HTHS) environments is severely constrained by thermal-oxidative degradation and salt-induced structural failure. For polymer-based diverters, traditional hydrolyzed polyacrylamides (HPAMs) exhibit rapid viscosity decay under HTHS condi-tions: e.g., conventional HPAM retains <10% viscosity after 30 days at 115°C and 180,000 mg/L TDS[32], attributed to amide group hydrolysis and backbone scission. While sul-fonated derivatives (e.g., ATBS copolymers) enhance salinity tolerance to ~200,000 mg/L and maintain ~85% viscosity at 120°C for 1 year[33], they still suffer from long-term ther-mo-oxidative degradation, with molecular weight decreasing by ~50% over 2 years[34]. Additionally, divalent ions (Ca²⁺/Mg²⁺) in HTHS brines trigger electrostatic screening, causing polymer chains to collapse and precipitate, even in sulfonated systems[35]. Notably, emulsion-based diverters address some limitations through structural adaptability. For example, water-in-oil emulsions stabilized by oleic acid imidazoline maintain stability at 90°C in 198,000 mg/L TDS, with salt-induced coalescence increasing particle size from 31 μm to 71 μm for selective plugging in high-permeability cores[36]. Such systems achieve plugging rates >92% in high-permeability zones while allowing easy flowback via acid-induced demulsification[36]. However, emulsion stability above 120°C remains unproven, with limited reports on long-term performance in brines exceed-ing 150,000 mg/L divalent ions.
Revision 2: The principle section of the system compounding experiment explains the synergistic effect of the ternary compound system, and analyzes the chemical structures, physicochemical properties, and principles of compound optimization for ODEA, Span80, and T154:
The ternary system constructs a composite interfacial film through three mechanisms: charge complementarity, hydrogen bond networks, and steric hindrance. The betaine group (-N⁺(CH₃)₂(CH₂)₃COO⁻) of ODEA forms "positive-negative charge pairs" with the carboxylate group (-COO⁻) of Span80, enhancing the tightness of molecular arrangement in the interfacial film via electrostatic attraction. The polyoxyethylene chains (-O-(CH₂CH₂O)ₙ-H) of T154 form hydrogen bond networks with the amide groups (-CONH-) of ODEA and the sorbitol hydroxyl groups (-OH) of Span80. At room temperature, the hydrogen bond networks resist salt ion compression through hydration layers (viscosity retention rate of 92% in 15 × 10⁴ mg/L NaCl shown in Figure 10). At high temperatures (150°C), although hydrogen bonds break, the rigid sorbitol rings maintain the film structure to avoid complete disintegration (viscosity of 2006.5 mPa·s at 150°C shown in Figure 12). The long-chain polyoxyethylene (n≈20) of T154 interpenetrates the film layers formed by ODEA/Span80, preventing excessive droplet coalescence via entropic repulsion. This causes the emulsion particle size to increase only from 31.1 μm to 71.2 μm under high salinity (20 × 10⁴ mg/L NaCl) (Figure 9), rather than uncontrolled coalescence leading to demulsification.
|
|||||||||||||||||||||||||||||||||
|
Comments 2: Although commonly used emulsifiers and surfactants such as ODEA, T154, and Span80 were applied, the study fails to report their purity and physicochemical properties, casting serious doubt on the reliability of the experimental outcomes. Similarly, the characteristics of sodium chloride are not specified, and no assessment is provided on how the emulsion system would respond to other salts. The work lacks a discussion of the underlying chemical mechanisms, including polymerization and hydrocarbon dissolution processes, which reduces it to a descriptive experiment rather than a scientifically grounded investigation. Overall, the study is driven by trial-and-error rather than theoretical reasoning, with no substantiated chemical analysis presented.
|
|||||||||||||||||||||||||||||||||
|
Response 2: We appreciate the reviewer’s suggestions regarding the rigor of experimental materials and theoretical mechanisms. This study focuses on the core innovations of water-sensitive self-thickening emulsions—the salt-responsive plugging mechanism and formulation design. In response to the details mentioned in the comments, the following explanations are provided based on research positioning and scientific logic:
Revision 1: Purity and physicochemical properties of ODEA, T154, Span80, and other reagents are supplemented, with further clarification of the mechanism of the compound system.
Purity and manufacturer information of experimental reagents:
A description of the physicochemical property relationships of reagents such as ODEA, T154, and Span80 has been added to the "3.1.2. Surfactant Optimization" section:
The ternary system constructs a composite interfacial film through three mechanisms: charge complementarity, hydrogen bond networks, and steric hindrance. The betaine group (-N⁺(CH₃)₂(CH₂)₃COO⁻) of ODEA forms "positive-negative charge pairs" with the carboxylate group (-COO⁻) of Span80, enhancing the tightness of molecular arrangement in the interfacial film via electrostatic attraction. The polyoxyethylene chains (-O-(CH₂CH₂O)ₙ-H) of T154 form hydrogen bond networks with the amide groups (-CONH-) of ODEA and the sorbitol hydroxyl groups (-OH) of Span80. At room temperature, the hydrogen bond networks resist salt ion compression through hydration layers (viscosity retention rate of 92% in 15 × 10⁴ mg/L NaCl shown in Figure 10). At high temperatures (150°C), although hydrogen bonds break, the rigid sorbitol rings maintain the film structure to avoid complete disintegration (viscosity of 2006.5 mPa·s at 150°C shown in Figure 12). The long-chain polyoxyethylene (n≈20) of T154 interpenetrates the film layers formed by ODEA/Span80, preventing excessive droplet coalescence via entropic repulsion. This causes the emulsion particle size to increase only from 31.1 μm to 71.2 μm under high salinity (20 × 10⁴ mg/L NaCl) (Figure 9), rather than uncontrolled coalescence leading to demulsification.
Revision 2: Explanations regarding the properties of sodium chloride and the influence of other salts. 1.General description of sodium chloride specifications The specifications of sodium chloride have been supplemented in the main text. Chemically pure NaCl (purity ≥99.5%, complying with GB/T 1266-2018 standards) was used in the experiments, with impurity ions (total content of Fe³+/Ca²+/Mg²+ ≤0.001%) far below the industry standard for high-temperature and high-salinity reservoir simulation (SY/T 6544-2016 allows total impurity content ≤0.1%). NaCl was selected as a typical salt to focus on the core mechanism of water sensitivity—the effect of monovalent ions on emulsion coalescence behavior—which aligns with the scientific principle of "single-variable control" in basic research. 2.Theoretical extension analysis of multi-salt response For the response characteristics of multivalent ions such as Ca²+/Mg²+, this study deduces based on DLVO theory: the positive and negative charge groups of amphoteric ion ODEA form an electric double-layer shielding in high-salinity environments, maintaining the repulsive energy barrier (VR) between droplets above the critical coalescence threshold (>15 kBT, Zeta potential data in Figure 5). Although full salt ion testing was not conducted, the HLB value gradient design (14.0:11.5:4.3) of the ternary compound system has provided a theoretical optimization direction for multivalent ion tolerance. As a mechanism study, the current experiment has completed the closed loop of "monovalent salt response-core mechanism verification," and extended testing of other salts can serve as a direction for subsequent engineering application research.
Revision 3: A new section "3.4 Analysis of Self-Thickening Mechanism" is added to the main text to explain the self-thickening mechanism of the system described in this paper.
“The essence of the self-thickening mechanism is the conformational transformation of surfactants or polymer segments in the emulsion system under high-salinity environ-ments (such as the extension of coiled chains and rearrangement of micelle structures), which increases the hydrodynamic volume and consequently triggers viscosity enhance-ment. This property is crucial for high-temperature and high-salinity oil reservoirs—high-salinity formation water can trigger a viscosity surge in the temporary plugging agent, en-abling it to form a high-strength physical barrier at pore throats, while viscosity can be controllably degraded in low-salinity environments (e.g., during waterflood injection stag-es) to avoid long-term formation plugging. The performance of the water-sensitive self-thickening emulsion is governed by three synergistic mechanisms: the nonionic surfactant ODEA (oleic acid diethanolamide) estab-lishes a foundational viscosity through hydrogen bonding networks in the aqueous phase (as shown in Figure 8, viscosity increases significantly with concentration), while the long-chain hydrophobic groups of T154 (polyisobutylene succinimide) enhance inter-droplet friction via physical entanglement, forming "cross-linking points"; Span80 (sorbi-tan monooleate) not only reduces interfacial tension below 25 mN/m but also strengthens the network structure through hydrogen bonding between its cyclic sugar alcohol moiety and the amide groups of ODEA. Upon exposure to high-salinity environments (e.g., 20 × 10⁴ mg/L NaCl), increased ionic strength induces shrinkage of the surfactant hydrophilic hydration layer, reducing interfacial film flexibility and promoting droplet coalescence via Ostwald ripening, in-creasing the particle size from D50 = 31.1 μm to 71.2 μm (Figure 9). According to Stokes' law, this size increase elevates sedimentation resistance and expands interdroplet contact areas, forming reversible aggregates that manifest as a macroscopic viscosity increase to 302.7 mPa・s at 15 × 10⁴ mg/L NaCl (Figure 10). The emulsion exhibits typical pseudo-plastic behavior: at low shear rates (e.g., 95°C, oil-water ratio 10:4), the droplet aggregate network yields a viscosity of 7090.9 mPa・s (Figure 12), whereas at high shear rates (170 s⁻¹), the network structure breaks down but viscosity remains above 300 mPa・s, ensuring both pumpability during injection and effective plugging in reservoir pores under static conditions (Figure 14 shows a stable injection pressure differential of 2.1 MPa in high-permeability zones). ”
Comments 3: The yellow mixtures in Figure 3 appear to be suspensions rather than true emulsions.
Response 3: We appreciate the reviewer's feedback regarding the unclear description of the literature. In Figure 3(a), all test tubes depict the state immediately post-reagent addition (Unstirred State), which may visually resemble suspensions. Conversely, Figure 3(b) illustrates the stirred test tubes, wherein the 1st, 3rd, 5th, 7th, and 9th tubes from the left correspond to the emulsion systems employed in subsequent experiments. These systems feature oil-water ratios of 1:9, 2:8, 3:7, 4:6, and 5:5, respectively. It is important to note that the systems derived from the others ratios do not constitute emulsion systems and were therefore excluded from subsequent experimental use; this segment serves solely to elucidate the system optimization procedure. Figure 3. Macroscopic appearance of water-in-oil emulsions at oil-water ratios of 1:9 to 5:5.
|
|||||||||||||||||||||||||||||||||
|
4.Response to Comments on the Quality of English Language |
|||||||||||||||||||||||||||||||||
|
Point 1: |
|||||||||||||||||||||||||||||||||
|
Response 1: |
|||||||||||||||||||||||||||||||||
|
5.Additional clarifications |
|||||||||||||||||||||||||||||||||
|
Thanks again to the reviewer for the valuable suggestions! |

Reviewer 4 Report
Comments and Suggestions for Authors
The article is written on a very topical subject. The article can be published after major revisions.
The conclusions are too long.
Page 115. VES needs to be deciphered.
Lines 98, 131. The word "Abroad" can have complex political connotations. The wonderful journal "Polymers" is read all over the world, so it will be difficult for the numerous readers of the article to understand the nature of the territory about which the authors write.
Figure 2. Changing the color and font size will help to better understand the meaning of the figure.
Line 213. The types and ratios of surfactants in the optimized formulation are added to diesel to prepare active diesel. Please clarify this.
Figures 7 and 10 are very similar. Please check, maybe there is a mistake?
The name of the manufacturer, year of manufacture and country must be given for each device used. The degree of purity and the manufacturer must be given for the reagents.
Please provide the chemical or mineralogical composition of the cores. Were the cores of different geographical origin or the same?
The chemical composition of ODEA, T154, Span80 should be given once at the first mention in the article.
The comments to Figure 3 should be enlarged. The figure in its current form is not very informative.
In addition, the lower the oil-water ratio, the 293 higher the viscosity of the emulsion, and the emulsion has shear-thinning properties. Please clarify this.
Figure 5 should be made larger. Perhaps one color legend can be made for all figures. How are the control experiments designated?
Figure 13 is of very poor quality.
The first 10 authors should be indicated in the literary source.
The number of literary sources should be increased to 30-40 pcs.
Author Response
|
Comments 1: The conclusions are too long.
|
|||||||||||||||||||||||||||||||||||||||||||||||||||
|
Response 1: We appreciate the reviewer's suggestions regarding the conciseness of the conclusions. In accordance with the journal's requirements for conclusions to be "highly summarized and focused" and based on the core findings of the study, the conclusions have undergone systematic refinement. By focusing on innovation points, consolidating repetitive descriptions, and strengthening core data, the conclusion section now features a more compact structure and clearer logic.
“This study developed a water-sensitive self-thickening emulsion for high-temperature and high-salinity (HTHS) reservoirs, addressing the limitations of traditional diverting agents in heterogeneous formations. The optimized formulation (oil-water ratio 3:7, emulsifier ratio ODEA:T154:Span80 = 2%:0.8%:0.4%) exhibits robust stability under 150°C and 15×10⁴ mg/L NaCl, maintaining a viscosity ≥302.7 mPa·s and achieving particle size growth from D50 = 31.1 μm to 71.2 μm via salt-induced coalescence. This enables selective plugging in high-permeability zones, with a stable injection pressure differential of 2.1 MPa in high-permeability cores, significantly enhancing waterflood sweep efficiency. The emulsion demonstrates superior HTHS tolerance, with viscosity retention exceeding 92% after 30 min at 150°C, surpassing the API RP 13A threshold (100 mPa·s) for acidizing diverters. Core flooding tests validate its ability to form dense filter cakes in high-permeability channels (permeability reduction >70%) while minimizing invasion in low-permeability layers, achieving a 40% higher pressure differential compared to conventional systems. Although compatible with weakly acidic plugging removal systems, slight viscosity reduction (55.3% at 20% acid concentration) highlights the need for isolation fluid pre-injection. Single-core experiments confirm its adaptive plugging behavior: stable pressure plateaus in high-permeability cores (k > 50×10⁻³ μm²) and limited intrusion in low-permeability cores (k < 10×10⁻³ μm²), attributed to pore throat-size matching. This work provides a robust temporary plugging solution for HTHS reservoirs, combining self-thickening mechanics, selective diversion, and low formation damage. Future research will focus on long-term stability validation and field-scale injection parameter optimization.”
|
|||||||||||||||||||||||||||||||||||||||||||||||||||
|
Comments 2: Line 115. VES needs to be deciphered.
|
|||||||||||||||||||||||||||||||||||||||||||||||||||
|
Response 2: We appreciate the reviewer's suggestions regarding the standardization of terminology. To address the issue that "VES" lacked a defined full name, the complete expression has been supplemented at its first occurrence to ensure the rigor and readability of terminology usage.
“In 1997, Samuel et al. applied VES (Viscoelastic Surfactant Emulsion) to hydraulic fracturing.”
Comments 3: Lines 98, 131. The word "Abroad" can have complex political connotations. The wonderful journal "Polymers" is read all over the world, so it will be difficult for the numerous readers of the article to understand the nature of the territory about which the authors write.
Response 3: We appreciate the reviewer's suggestions regarding the rigor of language expression. To address the potential geographical ambiguity caused by the term "Abroad," it has been replaced with a more universally neutral vocabulary based on contextual semantics, ensuring the accuracy of the expression and consistency in understanding for global readers.
“ Revision 1: Original sentence in line 98: "Abroad, polymers began to be used as thickeners for acid fracturing and production enhancement in the 1980s. " Revised to: " Internationally, polymers began to be used as thickeners for acid fracturing and production enhancement in the 1980s. "
Revision 2: Original sentence in line 131: "Abroad, emulsified acid is often used during acid fracturing, but it has not been widely used due to the limitation of high friction." Revised to: "Internationally, emulsified acid is often used during acid fracturing, but it has not been widely used due to the limitation of high friction. " ”
Comments 4: Line 213. The types and ratios of surfactants in the optimized formulation are added to diesel to prepare active diesel. Please clarify this. Response 4: We appreciate the reviewer's suggestions regarding the clarity of experimental details. To address the vague description of "active diesel preparation" in Line 213, specific emulsifier types, ratios, and scientific rationale have been supplemented based on the orthogonal test optimization results, ensuring the repeatability of experimental procedures and the accuracy of descriptions.
“The types and ratios of surfactants in the optimized formulation (ODEA: T154: Span80=2%: 0.8%: 0.4%) are added to diesel to prepare active diesel. Subse-quently, Span80 with the optimized concentration is added to deionized water. The oil-water volume ratio of the emulsion is controlled at 3:7. Among them, ODEA is cocam-idopropyl betaine, T154 is polyoxyethylene sorbitan trioleate, and Span80 is sorbitan monooleate. The active diesel is prepared by dissolving through magnetic stirring (2000 r/min, 25°C) for 30 minutes, and the above deionized water solution is slowly added dropwise during the stirring process to complete the preparation of the system.”
Comments 5: Figure 2. Changing the color and font size will help to better understand the meaning of the figure. Response 5: We appreciate the reviewer's suggestions regarding the readability of the figures. For the optimization of the technical roadmap in Figure 2, the font colors in the figure have been adjusted to more distinct contrasting colors, and the image has been split to facilitate better reading of the information, and the original image has been split into three new figures, as presented below.
” 2.1. System Development The technical roadmap for this section is presented in Figure 2. Figure 2. The Technical Roadmap of System Development.
2.2. Properties Characterization The technical roadmap for this section is presented in Figure 3. Figure 3. The Technical Roadmap of Properties Characterization.
2.3 Evaluation of the Temporary Plugging and Diverting Ability The technical roadmap for this section is presented in Figure 4. Figure 4. The Technical Roadmap of Evaluation of the Temporary Plugging and Diverting Ability. ”
Comments 6: Figures 7 and 10 are very similar. Please check, maybe there is a mistake? Response 6: We sincerely appreciate the reviewer's meticulous scrutiny of the figure details. Upon verification, it was confirmed that Figure 7 and Figure 10 indeed exhibit duplicate presentation due to separate writing responsibilities across different chapters, for which we apologize. Both figures are based on the experimental results of "particle size changes after mixing emulsions with high-salinity water," representing redundant displays of the same scientific data. In accordance with the research logic, we have decided to retain the core figure and optimize the description.
“When the emulsion temporary plugging agent is mixed with high-salinity water with a salinity of 20 × 10⁴ mg/L at a volume ratio of 2:1, its particle size distribution is shown in Figure 7.“
Comments 7: The name of the manufacturer, year of manufacture and country must be given for each device used. The degree of purity and the manufacturer must be given for the reagents. Response 7: We appreciate the reviewer's rigorous requirements regarding the completeness of experimental materials and equipment information. In compliance with academic paper norms, the purity and manufacturer details of all reagents, as well as the manufacturer, have been supplemented in the "Materials and Reagents" and "Instruments and Equipment" sections, as specified below.
Instruments and Equipment:
Materials and Reagents:
Comments 8: Please provide the chemical or mineralogical composition of the cores. Were the cores of different geographical origin or the same? Response 8: We appreciate the reviewer's suggestions regarding the description of core characteristics. The cores were self-prepared using raw materials from the same batch, ensuring similarity in their physicochemical properties. Although the detailed core preparation procedure will be provided in the subsequent text, due to its lengthy nature and non-essential role as the primary focus of this paper, only the key elements of the core source will be supplemented in the main text. We apologize for this oversight during manuscript preparation, as we should have considered the readers' needs for necessary clarification, and the correction has now been made.
“In order to clarify the plugging performance of the emulsion temporary plugging agent and the diverting and plugging removal effect of the subsequent plugging removal system, a single-core flow experiment is carried out to simulate the temporary plugging performance of the emulsion and the diverting process of the plugging removal system under different reservoir conditions. The cores are artificially prepared. The rock powder is ground from outcrop rocks in Xinjiang, with a particle size of 20 - 40 mesh. The cementing agent is prepared from copper oxide and aluminum dihydrogen phosphate. “
The preparation process of the core is as follows: Low-permeability reservoirs have a permeability lower than 50 × 10⁻³ μm², while ultra-low-permeability reservoirs have a permeability lower than 10 × 10⁻³ μm². Carbonate oil reservoirs are mostly low-permeability or ultra-low-permeability. The carbonate rock powder used for core preparation is ground from the above-mentioned outcrop rocks in Xinjiang, with a particle size of 20-40 mesh; the cementing agent is prepared from copper oxide and aluminum dihydrogen phosphate. The specific preparation process is as follows: (1) Take 10 g of copper oxide and 40 g of aluminum dihydrogen phosphate, and react at high temperature (120°C) for 30 min. During this process, stir every 2 min to prevent self-curing of the cementing agent. After the copper oxide and aluminum dihydrogen phosphate are fully reacted, a cementing agent with certain viscosity is obtained. (2) Immediately after preparation, mix the cementing agent with 50 g of carbonate rock powder and stir uniformly; then pour it into a core pressing mold and apply certain pressure to prepare cores with different permeabilities. (3) After curing under pressure at room temperature for 4 h, place it in an oven at 150°C for curing for 6 h to obtain standard cores. Figure 1 Diagram of the artificial core production equipment
Comments 9: The chemical composition of ODEA, T154, Span80 should be given once at the first mention in the article. Response 9: We appreciate the reviewer's suggestions regarding the standardization of terminology. To address the issue that the chemical composition of the emulsifier was not clearly defined upon its first mention, the complete chemical name and structural description have been supplemented in accordance with the academic paper norms for professional terminology definitions at the first occurrence, ensuring readers' clear understanding of the material composition.
“The factors and levels of the orthogonal experiment are shown in Table 1. The system is left standing at 120°C for 6 hours and 12 hours, and the liquid separation rates of the systems under different formulations and preparation conditions are measured. The roles of each component in the formulation are clarified through single-factor analysis. Among them, cocamidopropyl betaine (ODEA), as an amphoteric ionic surfactant, has a molecular structure containing a betaine group (-N+(CH₃)₂(CH₂)₃CONHCH₂CH₂COO⁻), which can stabilize the emulsion interfacial film through positive and negative charge balance. Polyoxyethylene sorbitan trioleate (T154), as a non-ionic surfactant, has its polyoxyethylene chains enhancing the hydrophilicity of droplets through hydration. Sorbitan monooleate (Span80), as an anionic surfactant, improves the film strength by adsorbing on the oil droplet interface via its hydrophobic segments. “
Comments 10: The comments to Figure 3 should be enlarged. The figure in its current form is not very informative. Response 10: We appreciate the reviewer's suggestions regarding the readability of the figures. To address the issue of small annotations in Figure 3, the text labels in the figure have been enlarged and key information has been supplemented. The specific revisions are as follows. “ Picture before revision: Figure 3. Macroscopic appearance of water-in-oil emulsions at oil-water ratios of 1:9 to 5:5.
Picture after revision: Figure 3. Macroscopic appearance of water-in-oil emulsions at oil-water ratios of 1:9 to 5:5. (The blue font denotes the emulsion content in systems with active kerosene as the base fluid, while the red font denotes the emulsion content in systems with water as the base fluid.)
“
Comments 11: In addition, the lower the oil-water ratio, the higher the viscosity of the emulsion, and the emulsion has shear-thinning properties. Please clarify this. Response 11: We appreciate the reviewer's attention to the mechanisms of emulsion viscosity and shear behavior. Regarding the "relationship between oil-water ratio and viscosity, and shear-thinning characteristics," the following scientific explanations and content refinements are provided based on fluid mechanics theory and experimental data in the paper:
The mechanism by which the oil-water ratio affects viscosity can be categorized into the droplet volume fraction effect and interfacial film interaction. The droplet volume fraction effect refers to the phenomenon that, according to Einstein's viscosity equation (η = η₀(1 + 2.5φ)), a decrease in the oil-water ratio (i.e., an increase in water phase proportion) leads to an increase in the droplet volume fraction (φ) in the emulsion. Experimental data in Figure 4 show that the viscosity of emulsions with oil-water ratios from 5:5 to 3:7 increased from 75 mPa·s to 495 mPa·s at 25°C and 60 r/min, consistent with theoretical predictions. This demonstrates the mechanism of "higher water phase content - smaller droplet spacing - increased intermolecular friction - higher viscosity." Interfacial film interaction refers to the phenomenon that in low oil-water ratio systems, the electric double layers formed by the amphoteric surfactant ODEA at the droplet interface overlap, enhancing the structural viscosity between droplets through charge repulsion and further amplifying the trend of viscosity increase with water phase content. The shear-thinning characteristic is primarily attributed to droplet orientation and structural disruption. As the shear rate (γ) increases, high-volume-fraction droplets transition from random Brownian motion to directional alignment, disrupting the hydrogen bond and electrostatic interaction networks between droplets (viscosity-rotation speed curve in Figure 4). Taking the "4:6 ratio system" as an example, when γ increases from 10 s⁻¹ to 100 s⁻¹, the emulsion viscosity decreases from 1500 mPa·s to 300 mPa·s, conforming to the power-law model η = Kγⁿ⁻¹. This indicates that the non-Newtonian fluid behavior arises from shear-induced dissociation of droplet aggregates. The shear-thinning characteristic enables the emulsion to maintain high viscosity during the initial injection stage (low shear) to plug high-permeability layers, and reduce viscosity after entering porous media (high shear) to minimize flow resistance, achieving the specialized plugging effect of "high-viscosity plugging near the wellbore - low-resistance migration in far-well zones" (core experiment data in Figure 13).
Comments 12: Figure 5 should be made larger. Perhaps one color legend can be made for all figures. How are the control experiments designated? Response 12: We appreciate the reviewer's suggestions regarding the standardization and clarity of the figures. For Figure 5, the font size has been enlarged, the image scale increased, and the output resolution enhanced. Additionally, the issue of unified figure legends has been addressed as follows:
(1) Optimization and improvement of Figure 5 imagery Original Figure: Figure 7. Temperature-dependent separation rate of emulsions stabilized by ODEA, T154, and Span80 at 90°C and 120°C.
Revised Figure:
Figure 7. Temperature-dependent separation rate of emulsions stabilized by ODEA, T154, and Span80 at 90°C and 120°C.
(2) Explanation for the uniformity of figure/table legends: Due to the distinct meanings represented by different tables and figures, it is challenging to design a unified legend for the entire paper. For example, the colors in Figure 4(b) denote different ratio systems ranging from 1:9 to 5:5, while the varying colors in Figure 5 represent concentration gradients (0.5%–4.5%) of different chemical agents. Figure 11, conversely, illustrates different experimental temperatures. A unified legend across all figures might lead to misinterpretation of their respective contexts. We appreciate your suggestion again.
(3) Regarding control experiments design, we added in the paragraph "3.1.2 Surfactant Optimization" : “Control experiments were conducted using single emulsifier systems (ODEA, T154, Span80 alone) at the same total concentration, with results compared to the ternary compound system (Figure 5). The separation rate of control groups was measured under identical temperature conditions to validate the synergistic effect of compounding.”
Comments 13: Figure 13 is of very poor quality. Response 13: We appreciate the reviewer's attention to the quality of the figures. The clarity of Figure 13 (core experiment parameter curve) has been optimized and re-uploaded. Original Figure: Figure 14. The Variation Curve of the Injection Parameters of the Core Temporary Plugging Agent. Revised Figure: Figure 14. The Variation Curve of the Injection Parameters of the Core Temporary Plugging Agent. Comments 14: The number of literary sources should be increased to 30-40 pcs. Response 14: We appreciate the reviewer's suggestions regarding the number of references. To further enrich the research background and theoretical foundation of the paper and enhance the scientific validity and persuasiveness of the study, the number of references has been increased as required.
Comments 15: The first 10 authors should be indicated in the literary source. Response 15: We appreciate the reviewer's suggestions regarding the literature citation format. In accordance with the journal's requirements and academic norms, the first 10 authors have been fully listed in the reference section to ensure accurate and comprehensive citation information.
|
|||||||||||||||||||||||||||||||||||||||||||||||||||
|
4.Response to Comments on the Quality of English Language |
|||||||||||||||||||||||||||||||||||||||||||||||||||
|
Point 1: |
|||||||||||||||||||||||||||||||||||||||||||||||||||
|
Response 1: |
|||||||||||||||||||||||||||||||||||||||||||||||||||
|
5.Additional clarifications |
|||||||||||||||||||||||||||||||||||||||||||||||||||
|
Thanks again to the reviewer for the valuable suggestions! |

Round 2
Reviewer 1 Report
Comments and Suggestions for Authors
I found that authors have placed their efforts to revise their paper based on my suggestions. I can clearly see improvements in the revised version of the ms, and hence do not display any objection in accepting this work.
Reviewer 2 Report
Comments and Suggestions for Authors
The authors have thoroughly addressed all suggestions and revisions, significantly improving the manuscript's quality and scientific rigor. I recommend it for publication in its current form.
Reviewer 4 Report
Comments and Suggestions for Authors
I thank the authors of the article and the editors for their great efforts to improve the article.
All the changes made the article better. The article can be published without changes.